# Deep history of cultural and linguistic evolution among Central African hunter-gatherers

Cecilia Padilla-Iglesias [1] ✉, Javier Blanco-Portillo [2], Bogdan Pricop [3], Alexander G. Ioannidis[4], Balthasar Bickel [3,5], Andrea Manica [6], Lucio Vinicius[1] & Andrea Bamberg Migliano [1,5] ✉

Human evolutionary history in Central Africa reflects a deep history of population connectivity. However, Central African hunter-gatherers (CAHGs) currently speak languages acquired from their neighbouring farmers. Hence it remains unclear which aspects of CAHG cultural diversity results from long-term evolution preceding agriculture and which reflect borrowing from farmers. On the basis of musical instruments, foraging tools, specialized vocabulary and genome-wide data from ten CAHG populations, we reveal evidence of large-scale cultural interconnectivity among CAHGs before and after the Bantu expansion. We also show that the distribution of hunter-gatherer musical instruments correlates with the oldest genomic segments in our sample predating farming. Music-related words are widely shared between western and eastern groups and likely precede the borrowing of Bantu languages. In contrast, subsistence tools are less frequently exchanged and may result from adaptation to local ecologies. We conclude that CAHG material culture and specialized lexicon reflect a long evolutionary history in Central Africa.

Recent fossil and genetic findings have revised the origins of *Homo sapiens* from less than 120,000 years[1] ago to almost half a million years ago[2–5]. Genomic analyses have revealed that some of the oldest human lineage divergences are represented by various extant African hunter-gatherer groups (that is, populations that primarily practise a foraging lifestyle for subsistence). Such groups include Khoisan-speaking hunter-gatherer groups in southern and eastern Africa[6–8]. In contrast, other studies have postulated that an encapsulation and isolation of Central African hunter-gatherers (CAHGs) from each other[9–11] and gradual intermixing with farming neighbours[12] would have resulted in a shallow cultural history[12–15]. Such a view has also been proposed based on their universal adoption of languages from neighbouring farmers, and therefore the absence of a common CAHG language or language family[16,17] (although see refs. 17,18). For example, the 10 CAHG groups in our study (7 western and 3 eastern groups) speak 13 languages from 3 highly differentiated linguistic families that are the product of farming expansions (Fig. 1 and Supplementary Tables 1–3).

Recent ecological, genetic and archaeological analyses have cast doubt on this interpretation, revealing various signs of long-term adaptation of CAHGs to current environments and independence from Bantu demographic history. For example, ref. 19 showed that ecological and climatic features of the Congo Basin can successfully predict 120,000 years of hunter-gatherer between-group interconnectivity, genetic exchange and continuous forest occupation. These results are in agreement with archaeological evidence[20], the deep genetic coalescence of CAHGs with other human lineages[3,21] and identification of

[1]Human Evolutionary Ecology Group, Institute of Evolutionary Anthropology, University of Zurich, Zurich, Switzerland. [2]Department for Biology, Stanford University, Stanford, CA, USA. [3]Department of Comparative Language Science, University of Zurich, Zurich, Switzerland. [4]Department of Biomedical Data Science, Stanford Medical School, Stanford, CA, USA. [5]Center for the Interdisciplinary Study of Language Evolution, University of Zurich, Zurich, Switzerland. [6]Department of Zoology, University of Cambridge, Cambridge, UK. ✉e-mail: cecilia.padillaiglesias@uzh.ch; andrea.migliano@uzh.ch

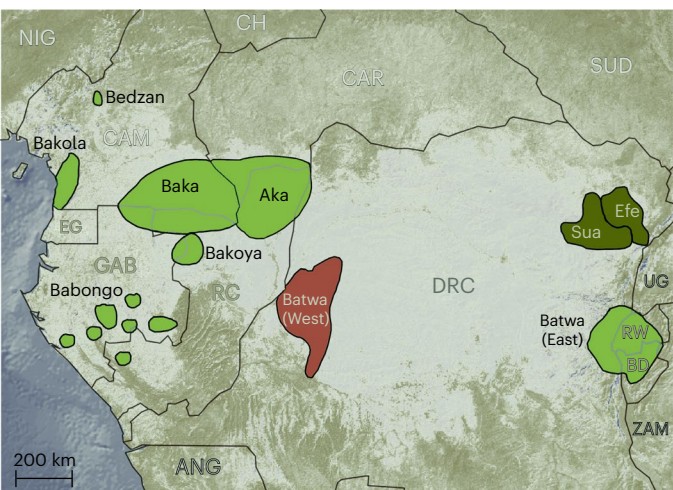

**Fig. 1 | Distribution of sampled CAHGs.** Genetic data were not available for the Western Batwa (shown in pink). Efe and Sua (dark green) were grouped into a single genetic unit (Mbuti). Cultural groups shown in light green correspond to a single genetic group. Abbreviated country names are indicated in capital letters. ANG, Angola; BD, Burundi; CAM, Cameroon; CAR, Central African Republic; CH, Chad; DRC, Democratic Republic of Congo; EG, Equatorial Guinea; GAB, Gabon; NIG, Nigeria; RC, Republic of Congo; RW, Rwanda; SUD, South Sudan; UG, Uganda; ZAM, Zambia. Map made with Natural Earth.

genes under selective pressure[21–25]. Finally, ethnographic studies have also questioned the isolation of CAHGs from one another, highlighting their widespread trips throughout the Congo Basin[26,27] to participate in rituals or search for spouses[27–29].

As proposed by the foraging niche hypothesis[30], a deep history of interpopulation connectivity across Africa would have various implications for our understanding of human origins and cultural diversity[31,32]. For example, rather than being the product of recent borrowing from farming groups, as is undoubtedly the case with language, other dimensions of CAHG culture such as musical instruments or subsistence tools may be the outcome of long-term, independent cultural evolution[12,33–35]. A long history of cultural exchange in eastern and southern Africa is exemplified by a system of bead transfers dated to 50,000 years ago[36] and by the long-distance trade of obsidian over 160 km nearly 200,000 years ago[37]. However, no study has investigated the structure of cultural diversity among hunter-gatherers in the large Central African region where CAHG groups may live as far as 2,000 km apart. Hence, showing the antiquity of cultural exchanges among CAHGs would provide key evidence for the role of long-term, continent-wide social networks in the cultural evolution of *H. sapiens*[4,30,38].

To investigate the possible effects of genetic history and group interconnectivity on cultural evolution in the Congo Basin, we integrated genetic data and cultural data on musical instruments, foraging tools and specialized vocabulary of CAHGs (Fig. 1). These three cultural domains are thought to be subject to different cultural evolutionary pressures due to the different functions they serve in society[32,39,40]. We compiled a comprehensive dataset of musical instrument repertoires, subsistence tools and their names in ten CAHG groups from ethnographies, photographs and museum collections (Fig. 1, Methods and Supplementary Dataset 1). We also obtained genome-wide single-nucleotide polymorphism (SNP) data from $N = 382$ individuals from 10 populations grouped into 9 units (Sua and Efe were grouped into a Mbuti group; Supplementary Table 4). We introduced a procedure for establishing the timing of cultural exchange events by performing genetic analyses of CAHG groups at three evolutionary depths: a full dataset incorporating segments resulting from Bantu introgression; a dataset including only genomic segments of CAHG origin, after masking away Bantu segments; and a set including only

genomic segments of deep CAHG ancestry not resulting from recent gene flow among CAHG populations. Our procedure shows that splitting genetic evidence into ancestry layers can reveal deep associations with cultural exchange and long-term interconnectivity among CAHGs independently from farming groups. We conclude that the early evolutionary history of CAHGs involved cultural exchanges across large-scale social networks and that the specialized cultural lexicons in CAHGs can provide insights into hunter-gatherer languages predating the expansion of farming groups in Central Africa.

## Results

### Musical instruments of CAHGs result from cultural exchange

We first asked whether cultural features of current CAHGs result from a deep history of evolution or were recently adopted from farmers in the same manner as their languages. We examined whether the distribution of CAHG musical instruments showed geographical patterning indicative of a shared genetic ancestry, or instead revealed stronger similarities between each CAHG group and its farming neighbours as a sign of recent cultural borrowing from farmers[41–44]. We calculated Jaccard distances between populations[45,46] based on the presence or absence of the 44 instrument types from our compiled dataset across 10 CAHG groups (mean of 14 instrument types per group; range between 10 and 25; Supplementary Fig. 1). The visualized results of a principal coordinate analysis (PCoA) using the NeighborNet algorithm revealed a clear geographical patterning, with defined western (Bedzan, Western Batwa, Bakola, Bakoya, Aka, Baka and Babongo) and eastern groups (Efe, Sua and Eastern Batwa) (Fig. 2a and Supplementary Figs. 2 and 3). Mantel tests and congruence among distance matrices (CADM) analyses provided statistical confirmation of the pattern, revealing a significant correlation between geography and music (Mantel statistic $\rho = 0.408$, adjusted $P = 0.024$; see Supplementary Table 5 for multiple matrix regressions and Supplementary Table 6 for CADM).

Next we assessed a possible link between the distribution of musical instruments and genetic affinities between CAHG populations. We began by examining whether CAHG genetic data also showed similar signs of geographical substructuring and differentiation independent of farmers. For this, we masked away Bantu-associated ancestry from all CAHG samples using a local ancestry inference method (Gnomix[47,48]). Following imputation for masked SNPs, multidimensional scaling (MDS; Methods) revealed affinities among CAHGs that largely mirrored geographical distances[49–51], with the first dimension segregating western from eastern CAHG groups and a second dimension separating northwestern and southwestern groups (Supplementary Fig. 4). Such spatial patterning was less clear when using either the unmasked dataset or the dataset including only genomic segments associated with Bantu ancestry (Supplementary Figs. 5–9). We found a significant correlation between genetic distances (pairwise fixation index ($F_{ST}$) measures exclusively based on CAHG ancestry genomic segments) and geographical proximity (Mantel statistic $\rho = 0.769$, $P = 0.001$; Supplementary Tables 6 and 7 for CADM results) even after controlling for ecological distance (measured as similarity in biome composition between CAHG territories) using multiple matrix regressions (Supplementary Figs. 10 and 11 and Supplementary Table 8). This was also revealed by our NeighborNet plot of between-group genetic distances (pairwise $F_{ST}$[52]) (Fig. 2b).

Having found evidence for the spatial structuring of both CAHG cultural and genetic diversity, we evaluated their potential associations. We implemented an isolation-by-distance model using SpaceMix[53] (Methods) whose null hypothesis is that the spatial distribution of CAHG genetic or cultural diversity reflects neutral diffusion among neighbouring populations, rather than substantial demic processes such as population replacement. We obtained a map in which mean group distances based on CAHG genetic pseudo-coordinates are smaller than their actual geographical distances (($t$-test with 80 degrees of freedom) $t(80) = 3.8849$, $P < 0.001$), indicating widespread gene flow across Central Africa and hence deviation from the null hypothesis (Supplementary

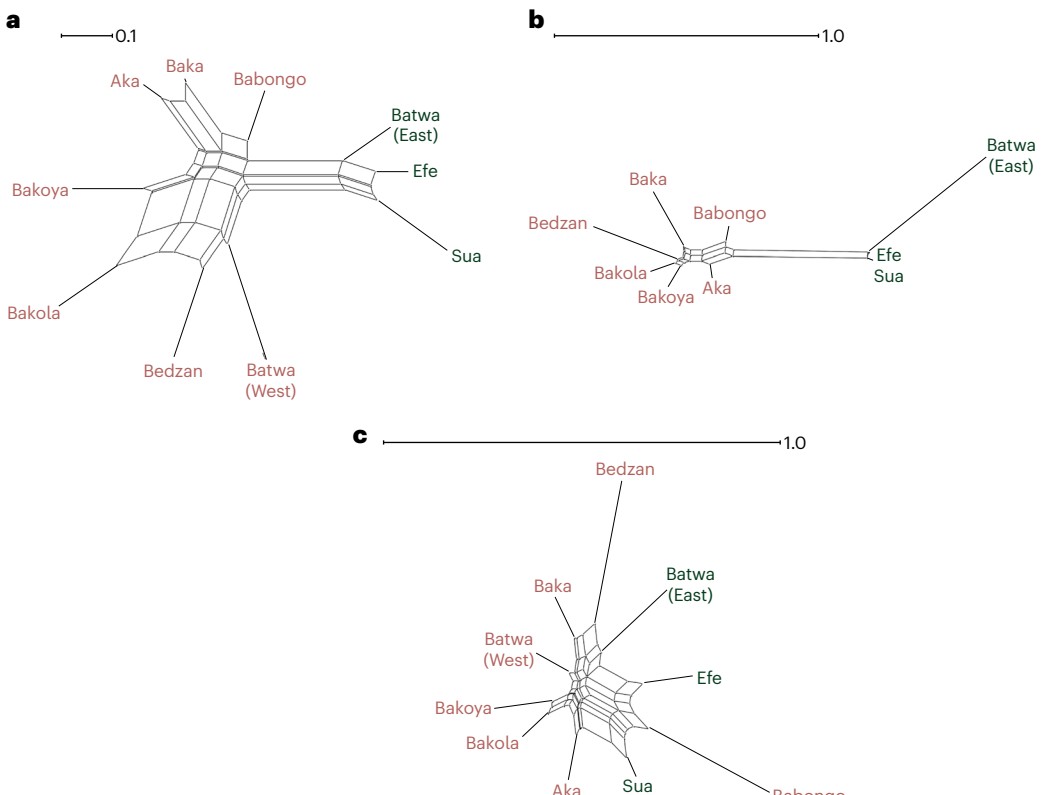

**Fig. 2 | Genetic and cultural distances between CAHGs. a–c,** NeighborNet graphs based on dimensionality reduced distance matrices for musical instruments (**a**), genes using exclusively CAHG ancestry segments (**b**) and subsistence tools (**c**) (see Methods for details). In pink, Western CAHG groups; in green Eastern CAHG groups. Scale bars indicate the distance from the Neighbournet joining algorithm in the original units of the data.

Fig. 12). SpaceMix analysis detected signs of recent gene flow even in the most geographically isolated populations, such as the Bedzan (12.8% of their genome originating in other CAHGs) or the Eastern Batwa (6.3%) (Supplementary Fig. 13 and Supplementary Table 9). Similarly, musical instrument pseudo-coordinates were closer to the actual CAHG geographical coordinates than to genetic pseudo-coordinates and also overall closer to one another than their actual geographical locations (Supplementary Fig. 12 and Supplementary Table 10). Overall our results suggest that demographic exchanges have led to the transfer of both genes and musical instruments among CAHG groups.

### Identical-by-descent segments reveal history of musical instrument exchange

We then investigated the time depth of cultural exchanges among CAHGs. Although the origin of CAHG musical instruments cannot be directly dated, they must be either a product of long-term evolution in Central Africa or only recently exchanged among CAHGs after farming populations expanded into the region and admixed with local hunter-gatherers. The length of shared identical-by-descent (IBD) fragments has been widely used for dating genetic exchanges of recent but nonetheless variable antiquity between populations[50,54–56], including CAHGs[19], with longer shared IBD blocks being the product of more recent admixture events (as fewer recombination events have taken place in admixed individuals)[57] (Supplementary Fig. 14). Therefore, we propose a method that searches for associations between cultural exchange and genetic events dated by IBD segments (Supplementary Text 1). This method is based on estimating correlations between musical instrument similarity and genetic similarity after splitting our CAHG genomic dataset into three levels: first, all genomic fragments, including those recently introgressed from Bantu populations; second, all fragments of CAHG ancestry (by masking away Bantu genomic segments); and, third, the remaining CAHG fragments after the exclusion of IBD fragments recently

exchanged among CAHGs. It should be noted that the last set contains both fragments of more ancient ancestry and others whose antiquity cannot be established (Fig. 3 and Supplementary Tables 11 and 12).

This approach implies various predictions: (1) if we only find a significant correlation between musical instrument and genetic similarity in the full genomic dataset, musical instruments have most likely been recently introgressed from neighbouring farmers, together with genes; (2) if we instead find correlations between similarity in musical instrument repertoires and genomic components exchanged among CAHGs, musical instruments are more likely to have been originated before the arrival of Bantu farmers, but continued to be exchanged among hunter-gatherers in recent times; and (3) if musical instruments only correlate with the remaining genomic fragments after the exclusion of IBD fragments exchanged among CAHGs, the distributions of both genetic fragments and musical instruments are likely to be the result of the prefarming shared history of CAHGs.

Mantel tests showed that CAHG genetic similarity based on the full dataset including Bantu fragments did not correlate with musical instrument similarity, a result also obtained when we used only Bantu fragments (Table 1; Supplementary Table 6 for CADM results). In contrast, genetic similarity based either on the set of all fragments of CAHG ancestry or on the set excluding IBD fragments recently shared between CAHGs shows significant and virtually identical correlations with musical instrument similarity. This suggests that genetic and cultural exchanges between hunter-gatherer groups are ancient and have remained mostly unaffected after recent farming expansions, pointing to a deep history of hunter-gatherer networks and uninterrupted cultural interconnectivity across the Congo Basin.

### Music lexicon predates the arrival of Bantu languages

Although CAHGs speak widely differentiated and mutually unintelligible languages recently acquired from farming groups (Supplementary

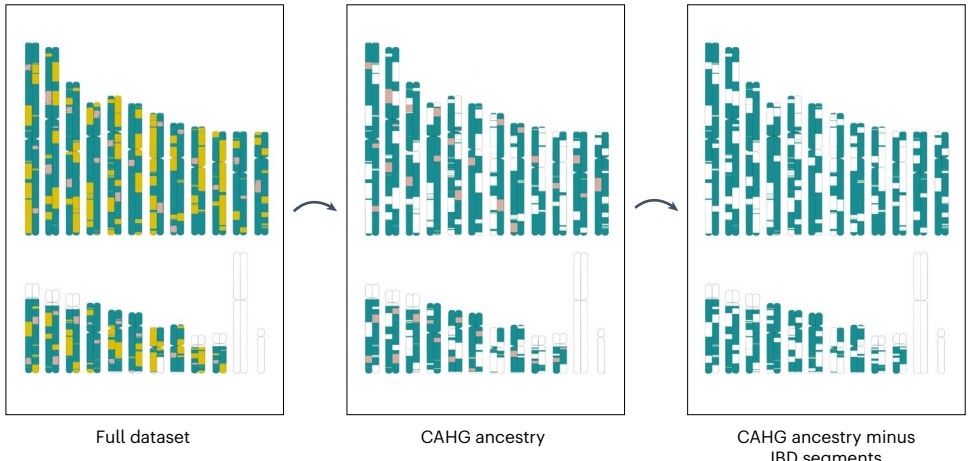

Full dataset CAHG ancestry CAHG ancestry minus IBD segments

**Fig. 3 | Fractional analysis of IBD fragments.** Schematic representation of the three genetic sets obtained through a two-step masking process on a human karyotype. The first set includes all genomic segments including segments of Bantu origin (yellow) and shared IBD segments between individuals from different CAHG populations (green and pink). The second set includes only genomic segments of CAHG ancestry (green and pink) after the removal of Bantu segments. The third set includes the remaining genomic segments of CAHG ancestry (green) after removing all IBD segments exceeding 1 cM shared between individuals from different CAHG populations.

**Table 1 | Mantel tests of correlations between cultural distances (musical instruments and subsistence tools) and geographic, ecological and genetic distances**

| | Musical instruments | | Subsistence tools | |
|---|---|---|---|---|
| | Mantel rho statistic | Adjusted *P* | Mantel rho statistic | Adjusted *P* |
| Geography | **0.408** | **0.024** | 0.207 | 0.152 |
| Ecology | 0.116 | 0.441 | **0.480** | **0.034** |
| Genes (full dataset) | 0.312 | 0.135 | 0.087 | 0.411 |
| Genes (CAHG ancestry) | **0.416** | **0.032** | 0.312 | 0.113 |
| Genes (CAHG ancestry without IBD segments) | **0.422** | **0.032** | 0.215 | 0.162 |
| Genes (Bantu ancestry) | −0.01 | 0.508 | −0.242 | 0.729 |

Genetic data were split into distinct ancestry levels: all genomic segments, segments of CAHG ancestry, and segments of CAHG ancestry excluding shared IBD segments between CAHG and segments of Bantu ancestry. *P* values were adjusted using the Benjamini–Hochberg procedure. Bold numbers represent significant statistics at an adjusted significance threshold of 0.05.

Figs. 15 and 16 and Supplementary Tables 2 and 3), our results suggest that their musical instruments have a more ancient origin. Hence, either CAHGs adapted words from farmer languages to name their instruments or they preserved the original names from their lost languages. We conducted an analysis of the specialized lexicon for musical instruments across CAHG groups to identify music-related words possibly predating the spread of farming groups and their languages. We compiled from primary sources a total of 183 words designating musical instruments from 10 groups (mean words per group = 18, range = 3–46). Despite having lost their original languages, CAHGs share 23 music-related words: 17 words shared between 2 groups, 3 among 3 groups, 3 among 4 groups and 1 among 5 groups (Supplementary Figs. 15 and 17, Supplementary Table 13, Supplementary Dataset 2 and Methods). Of those 23 words, 15 are exclusive to hunter-gatherer populations and not found among any of the farming neighbours of CAHGs (Fig. 4).

For example, the word *ngbídí* (see Supplementary Table 13 for variants) denotes a musical bow with two strings that is played exclusively by female CAHGs in the exact same manner by both eastern (Efe) and western (Aka and Baka) groups living over 2,000 km apart and not found in any other population[58]. Another shared word exclusive to CAHGs is *ngombi*, denoting the harp present in the Baka, Babongo, Aka and Batwa in the west (the latter being at least 400 km away from the other 3 groups). It is also relevant that Aka (Bantu C10 speakers) and Baka (Ubangian speakers) neighbours speak very distinct languages and yet share various unique words denoting exclusively shared musical instruments, such as *haka* (ankle rattles), *bogongo* (zither harp), *pole.pole* (seed whistle), *mobio* (flute) and *mokinda* (single-skinned drum). The presence of these instruments and words shared between different CAHG groups but not with their respective Bantu-speaking and Ubangian farming neighbours nor with other farming groups speaking related languages has previously been interpreted as evidence for their descent from a common ancient population[43,59].

To confirm that the sharing of words for musical instruments between populations was not the product of inheritance from the same or closely related farmer languages, we calculated linguistic distances between all pairs of CAHG languages using three methods. First, we used the Automated Similarity Judgment Program (ASJP) database to calculate pointwise mutual information (PMI) distances between all pairs of CAHG languages[60,61]. PMI has been shown to be informative for performing distance-based phylogenetic inference and it provided a measure of general vocabulary similarity between CAHG groups (Supplementary Fig. 16)[61,62]. We then calculated phylogenetic distances by counting the number of internal nodes between each pair of CAHG languages using the Glottolog trees[63]. For the subset of Bantu languages in our sample, we also obtained phylogenetic distances based on the most recent dated phylogeny of Bantu languages[64] (Methods). The correlation between PMI distances and Glottolog distances, as well as between PMI distances, and phylogenetic (patristic) distances were high (Spearman $\rho = 0.60$, $S = 6353.1$, $P < 0.001$; and Spearman $\rho = 0.86$, (Spearman's rank order correlation) $S = 2223.7$, $P < 0.001$, respectively). Our rationale was that if the sharing of words for musical instruments between CAHG groups results from their current languages being closely related, groups speaking more closely related languages should share more music-related words. If instead shared musical instrument words are (as with the musical instruments themselves) the product of a long cultural evolutionary history predating recent language replacement, genetically closer CAHG populations should share more words for musical instruments than linguistically related populations.

Zero-inflated Poisson models showed that genetic and geographical proximity were significant predictors of the number of shared musical instrument words between populations, whereas phylogenetic linguistic proximity and overall language similarity (that is, smaller

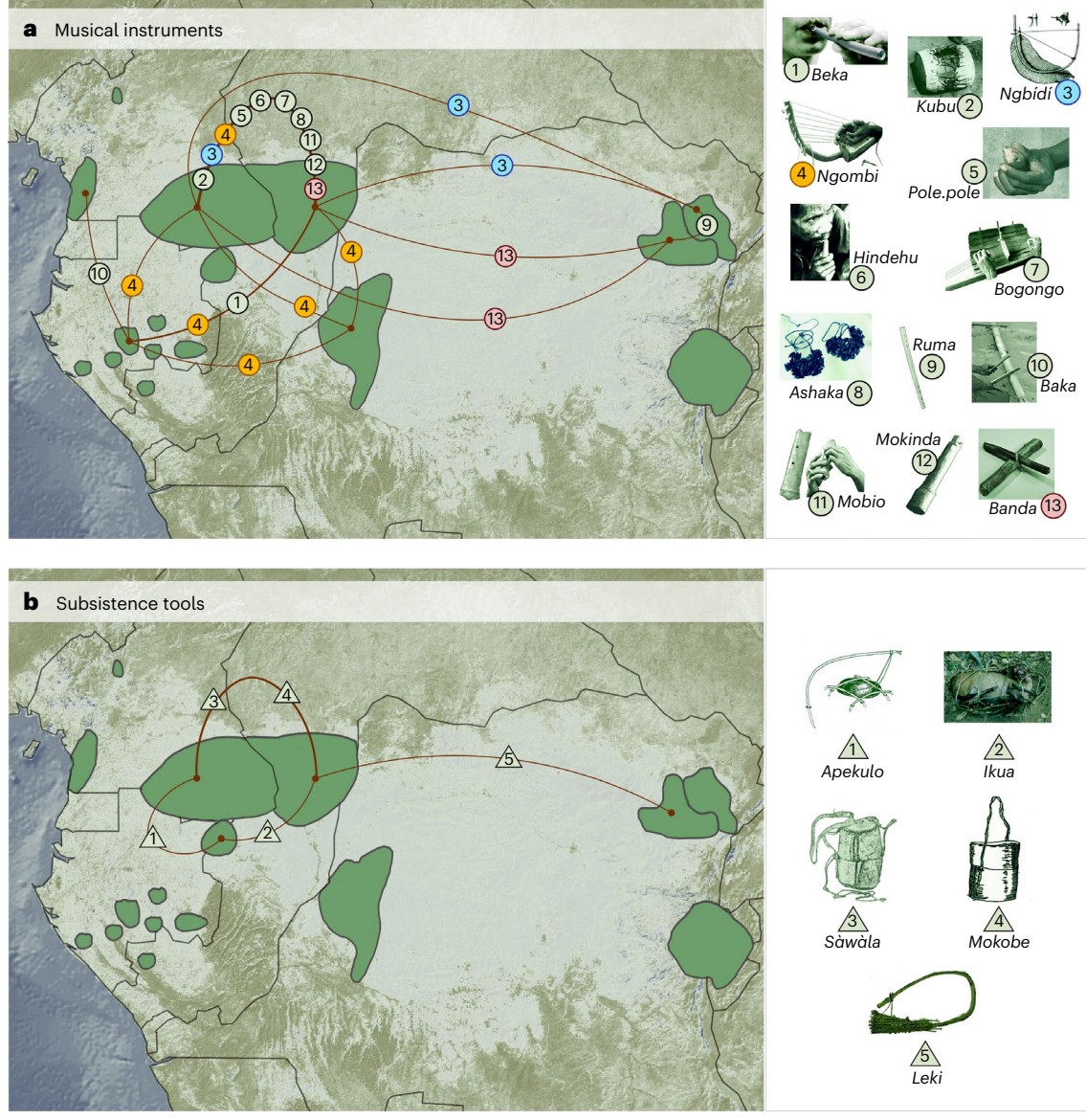

**Fig. 4 | Networks of specialized vocabulary for musical instruments and subsistence tools exclusive to CAHGs. a,b**, Shared vocabulary representing musical instruments (**a**) and subsistence tools (**b**) in CAHGs. Edge widths reflect the number of unique shared words between two populations. Each symbol (circles for musical instruments; triangles for subsistence tools) indicates one shared word between populations. Numbers inside symbols denote the corresponding musical instrument or subsistence tool. Symbol colours were added to facilitate visualization of instruments for which the corresponding word was shared between more than two groups. Map made with Natural Earth. Image credits for panel **a**: 1, 6, 10, 11, Africa Museum Musical Instrument collection, Tervuren; 2, 4, 5, 7–9, 12, 13, Susanne Fürniss; 3, ref. 43. Image credits for panel **b**: 1, 5, ref. 110; 2, Nikhil Chaudhury; 3, Africa Museum Musical Instrument collection, Tervuren; 4, ref. 18. All images redrawn by Rodolph Schlaepfer.

patristic, PMI or Glottolog distances) were not (Supplementary Tables 14–19). In fact, when considering patristic distances, we found the opposite to be true, with groups most distantly related linguistically sharing more musical instrument words (Supplementary Tables 16 and 19). This association between shared musical instrument lexicon with deep population history, but not with overall linguistic relationships or similarities, held both when considering all shared terms between CAHGs and when only considering those unique to CAHGs. For example, the two groups with the most similar general lexicons were the Aka and the Eastern Batwa (PMI = 0.71), which share no musical instrument words. In contrast, the Aka and the Baka are the genetically closest groups in our sample and share the greatest number of musical instrument words (N = 15, 12 of which are unique to CAHGs) but their general lexicons are substantially more distant (PMI = 0.81) than our sample average (PMI = 0.74).

Our findings also match popular legends in Central Africa describing a deep shared ancestry of musical instruments predating CAHG divergence and the common origins of the single-stringed musical bow *gongo* (shared across different CAHG groups) and of the CAHGs themselves[65,66]. Overall music-related words used by current CAHGs seem to represent words that originated in their extinct languages and the distribution of these words offers a glimpse at a shared linguistic ancestry across CAHGs[43,59].

## Subsistence tools represent adaptations to local environments

We also compiled complete subsistence tool repertoires from 10 CAHG groups (Methods), obtaining a total of 40 and average of 17 types of subsistence tool per group (range = 9–25; Supplementary Fig. 1). The distribution of tools among CAHG groups followed a distinct pattern

from that of musical instruments. First, the number of subsistence tools and musical instruments in groups were not correlated (Spearman $\rho = -0.073$, $S = 116.95$, $P = 0.839$). Mantel tests, multiple matrix regressions and CADM analyses identified no correlations between tool sharing and geographical distance (Table 1 and Supplementary Tables 5 and 7). SpaceMix analyses showed that the distance between the geographical coordinates of groups and their pseudo-coordinates estimated from subsistence tools was generally much greater (for 7 of 9 CAHG groups) than those between geographical coordinates and pseudo-coordinates estimated from musical instruments ((Wilcoxon signed-rank test statistic) $W = 20$, $P = 0.038$; Supplementary Fig. 18). As a result, the structure of spatial variation in subsistence tools showed a much worse match with the geographical distribution of groups (Supplementary Fig. 12). A NeighborNet visualization of PCoAs based on subsistence toolkit repertoires failed to identify the separation between western and eastern groups (Fig. 2c), between adjacent populations (Aka and Baka or Sua and Efe; Table 1 and Supplementary Fig. 2) or any clear geographical pattern. Finally, we found no correlation between tool similarity and genetic similarity when considering any of the three genomic sets (Table 1 and Supplementary Table 7). This indicates that diffusion of subsistence tools between groups has not regularly occurred during events of genetic exchange.

The only significant predictor of foraging tool distribution was similarity in ecology (biome composition; Table 1 and Supplementary Table 7 for CADM results). We noticed a nearly universal presence across populations of certain items, such as spears, hunting bows or arrows, intrinsically linked to the hunter-gatherer niche and therefore not expected to respond to ecological variation. However, the dataset also included more sparsely distributed items related to resource exploitation in specific ecological settings, such as hooks present only in the Baka and Bedzan, which are two geographically distant groups that are equally dependent on aquatic resources. The association with ecology but independence from demographic exchanges and cultural diffusion suggests that the distribution of such items may be more parsimoniously explained by convergent adaptation to similar environments.

Next we asked whether an ecology-driven pattern would also underlie the distribution of words for subsistence tools. From primary sources we identified 89 words designating subsistence tools in 9 populations (mean number of words per group = 9, range = 0–28; Supplementary Fig. 17 and Supplementary Table 13). In contrast to the more extensive sharing of musical terms, only ten tool names were shared between two groups and one name between three groups, whereas only three names denoted objects unique to CAHGs. We found that words designating musical instruments were more likely to be shared between CAHG groups than words designating subsistence tools ($W = 204$, $P < 0.001$) but the limited number of shared words related to tools precluded further statistical assessment. In summary, tool names are much less commonly shared than musical terms, even among groups sharing similar ecologies.

## Discussion

Our results provide evidence that long-term, large-scale social networks underlie a deep history of material cultural and lexical evolution among CAHGs. Such a deep history is one of the many social and evolutionary consequences of the high rates of between-group mobility evolved in hominin ancestors and is still observed in extant hunter-gatherers[30,67]. These consequences include the maintenance of high levels of cooperation in the absence of explicit punishment mechanisms[68] or the unique pattern of coresidence of unrelated individuals observed in hunter-gatherer camps[69]. At a regional scale, between-group mobility generates multilevel networks in which household clusters and camps are interconnected into larger, multicamp structures[70]. Such structures represent properties that facilitate cultural evolution and knowledge specialization through dedicated channels of social transmission[31,32,71].

Recently, we provided evidence for the long-term existence of such networks at a much larger scale, spanning the entire Central Africa for at least the last 120,000 years based on paleoclimatic reconstruction and archaeological data[19]. By combining cultural, linguistic and genetic data in the current study, we have shown that although some aspects of CAHG culture such as language are evolutionarily recent and borrowed from Bantu- and Ubangian-speaking groups, other domains reveal signs of long-term evolution[12,17]. Although extensive demographic and cultural exchanges with incoming farming populations undeniably took place over the last millennia, interconnectivity between CAHGs was not interrupted, thus preventing their isolation from each other[5,19,72,73]. For this reason, CAHG material culture is both the product of long-term contact with other hunter-gatherers and of recent exchanges with farming groups. This ability of maintaining large-scale social networks throughout the evolutionary history of our hunter-gatherer ancestors might represent a fundamental step in the divergence between hominins and our closest relatives[30].

The intricacy of CAHG cultural evolution is expressed in the distinct patterns of geographical distribution observed in musical instruments, subsistence tools and specialized lexicon. Musical instruments present a distribution suggesting exchanges over thousands of kilometres between western and eastern regions of the Congo Basin. We have introduced an analytical approach that decomposes genomic fragments into sets representing distinct ancestry depths and in this way we were able to show that the distribution of musical instruments across CAHGs is linked to genomic fragments not derived from recent exchanges with Bantu farmers. This suggests that splitting the genomic evidence into fractions of distinct antiquity can at least reject certain evolutionary and demographic scenarios, as we have exemplified above by ruling out a recent Bantu origin for musical instruments found in CAHG groups. Importantly, this approach can provide a clear link between the distribution of a cultural trait and an underlying pattern of demographic contact among populations.

As shown above, we identified an association between demographic exchanges among CAHGs with musical instruments not interrupted by contact with farming groups. This was not the case for subsistence tools, which suggests a possible long-term role of music in establishing cultural group identities. Music is an integral part of CAHG rituals that often differs among cultural groups and brings together individuals living far apart[28,29]. A common function of CAHG rituals is to allow members from distinct groups to meet unrelated cooperative or marriage partners, which may provide a social and mechanistic explanation for the correlation between diffusion of genes and musical instruments[28,29]. In fact, it has been reported that rituals in which the Aka and Baka use similar musical instruments are also named the same in both populations, such as the ritual to invoke the success of a quest (*zɔ̀bɔ̀kɔ̀*, *ndàmbò*, *è.sà* and *mbèlà*), to prepare for the death of an animal (*mò.nzòlì* and *kóbá*) or to celebrate the first capture of an important animal by a young man (*mò.póndí*)[42].

In contrast, the significant correlation between the distribution of subsistence tools and local ecological conditions provides strong evidence for long-term adaptation of CAHG groups to forest environments in the Congo Basin. The deep adaptive history of foraging tools may explain why ecological variables can successfully predict the distribution of both current and past hunter-gatherers in Central Africa[19]. A similar pattern was found among San hunter-gatherer groups across the Kalahari Desert, where shared environments could explain 90% of the variance in their projectile points[74].

We have also shown that although CAHGs have adopted languages from farmers, music-related words were most likely inherited from ancestral populations together with the corresponding musical instruments. In contrast, the fact that very few tool-related words are shared between CAHGs may reflect either a convergent origin of the similar tools in groups with divergent languages (and hence distinct words created for similar tools) or an ancient common ancestor to both

subsistence tools and tool-related words, with tool (but not word) differentiation over time being limited by ecological constraints. Overall the distribution of music-related words across Central Africa points to an ancient linguistic network interconnecting western and eastern populations[43]. We believe that extending our procedure may lead to the identification of other ancient shared linguistic terms in cultural domains known to precede the arrival of farming groups, such as vocabulary related to forest, forest life, rituals regarding the forest and kin relationships[18,66].

In summary, our results provide cultural, linguistic and genetic evidence that extant CAHGs are the outcome of a long history of between-group interconnectivity and occupation of forest environments in the Congo Basin. They also point to a key role played by extended social networks in circulating material culture, words and genes. Although current CAHGs have also maintained extensive contact with neighbouring farmers, we have shown that genetic and cultural evolution has taken place in Central Africa as a result of long-term and large-scale exchanges among CAHGs groups. Future studies should involve more comprehensive genetic, linguistic and cultural analyses including CAHGs alongside other African hunter-gatherers from southern and eastern Africa. Moreover, they should aim at including multivariate analyses of cultural and linguistic variation (beyond presence–absence) to pinpoint the drivers of biological and cultural evolution at a continental scale.

## Methods

### Material cultural corpus

We obtained data on musical instruments and subsistence-related tools (hunting, gathering and fishing) from $N = 115$ primary sources, including ethnographic accounts and photographic and video material specifically designed for documenting the cultural repertoires of a specific group in those different domains, and the ethnographic collection of the Royal Museum for Central Africa in Tervuren (Belgium) (Supplementary Dataset 1). Note that the latter comprised multiple primary sources. All the records were from the twentieth and twenty-first century. In all cases, we contrasted the data provided by every source with at least one other source and, in most instances, personally contacted the relevant ethnographers to confirm the presence or absence of each of the recorded objects. As different names are sometimes used to designate the same CAHG population, we only included data for which specific collection locations were indicated. When available, we also recorded (from the same source) the name or names used in the local language to designate each of the objects. We were able to obtain $N = 1,366$ material cultural records comprising full repertoires of musical instruments and subsistence items from 10 CAHG groups (Supplementary Table 1 and Supplementary Dataset 1). Overall we documented 44 different types of musical instrument and 40 different types of subsistence tool. At the same time, we compiled $N = 183$ different words to designate musical instruments from 10 different populations and $N = 89$ words to designate subsistence tools from 9 populations (see Supplementary Fig. 17, Supplementary Table 13, and Supplementary Datasets 1 and 2 for variation in terminology).

### Geographical location and territories

A geographical area was assigned to each of the cultural groups by georeferencing a previously published map[16,17] and spatial polygon objects were created from the territory occupied by each cultural group (Fig. 1). Given that groups inhabited relatively small areas, we took the coordinates for the centroid of each territory as the location for each cultural group[16,17].

### Genetic samples

Genetic data from CAHGs were obtained by merging 4 different publicly available datasets[24,75–77] comprising individuals from 13 African populations (Supplementary Table 4). Ethical approval for using data in

each of the datasets was granted by the relevant data access committees and signing officials for each of the publicly available data repositories (EGA and dbGaP) and by the Ethics Commission (Ethikkommission) of the University of Zurich (permit number 20.2.8). Each dataset was previously filtered for relatedness and Hardy–Weinberg equilibrium before we accessed the data. The data were merged using PLINK[78] and R[79] so that only SNPs common to all studies were retained. We removed multi-allelic loci, filtered data for relatedness >0.0886 with KING[80] and a random individual was chosen from each related pair. Using PLINK, the data were then filtered to remove SNPs with a minor allele frequency <0.05. A pruned version of the dataset was created with PLINK –indep-pairwise 50 5 0.2 to account for linkage disequilibrium.

The non-pruned dataset comprised 511 individuals and 431,867 SNPs from 13 different African populations and the pruned dataset comprised 511 individuals and 193,718 SNPs. For the purposes of this study, we merged BakaG and Baka as a single Baka population and BabongoE and BabongoS as a single Babongo population (Supplementary Table 4). The unpruned dataset was used for Gnomix, SpaceMix, all ancestry-specific analyses and for calculating runs of homozygosity. The pruned dataset was used for ADMIXTURE analyses[81]. All samples were analysed on the GRCh37 (hg19) genome build.

We recorded the geographical location from which each sample was obtained along with the cultural group to which it belonged (Supplementary Table 1). Note that the Mbuti genetic samples attributed to one population encompassed more than one cultural group (Efe and Sua). In this case, the two cultural groups were assumed to not be genetically differentiated from one another.

### Unsupervised ancestry clustering analyses

We performed unsupervised ancestry clustering using ADMIXTURE[81] on both our full pruned dataset and on a reduced subsample of our pruned dataset that included exclusively CAHG populations and a few unadmixed farming populations commonly used as references in population genomic analyses (Supplementary Table 4). ADMIXTURE estimates for every individual the proportions of the genome originating from $K$ ancestral populations, with $K$ specified a priori. The program was run at $K$ values from 1–8 with cross-validation error estimation, default values for fold iteration ($v = 5$) and a random seed for each $K$ value (Supplementary Figs. 19–21). Therefore, for each value of $K$, we ran as many iterations as required for the log-likelihood to increase by less than $\varepsilon = 10^{-4}$ between iterations ($\varepsilon$ is the difference in log likelihood; see Supplementary Table 20 for number of iterations required for each $K$ value). See Supplementary Table 21 for average ancestry proportions.

### Local ancestry inference and masking

We used Gnomix[47] on our dataset to perform semisupervised local ancestry inference with references from $K = 2$ ancestry clusters, 1 corresponding to CAHG ancestry components and the other to Bantu-associated agriculturalist ancestry. Gnomix has been shown to achieve higher accuracy than any other available local ancestry inference method[47], even when data at much higher resolutions are available[47]. Following refs. 24,82, reference panels for each ancestry cluster were created by selecting unadmixed agriculturalist and CAHG individuals (with >98% agriculturalist- or CAHG-associated ancestry) indicated by the unsupervised global ADMIXTURE $K = 2$ (ref. 81) clustering run described above, selecting a balanced number of individuals for each panel. The choice of reference samples was also corroborated by previous analyses of the same genomes. This resulted in a reference panel of $N = 31$ CAHG and $N = 38$ agriculturalist individuals. As recommended by the developers of the software, we used 15% of the data for training set 1, 80% for training set 2 and 5% for validation, a window size of 0.2 cM and an r_admixed parameter of 1 in Gnomix[47]. Samples were first phased by Beagle v.5.3 (ref. 83) with default settings. Across chromosomes, the classification accuracy for haplotypes of CAHG ancestry was 88% (range = 83–95%) and for

haplotypes of agriculturalist ancestry the classification accuracy was 84% (range = 76–88%) (see Supplementary Table 22 for the full confusion matrix). This classification accuracy is much higher than what is normally achieved by other local ancestry inference methods, even when data at much higher resolutions are available[47]. We removed Bantu-associated chromosomal segments in a procedure known as masking[84,85]: SNPs located in certain ancestry segments, as classified by Gnomix, were removed, or masked, from downstream analyses. We refer to the remaining unmasked chromosomal segments as CAHG chromosomal segments and we refer to analyses that use only these segments as CAHG ancestry-specific analyses.

To eliminate between-population shared long IBD segments, which are the product of recent genetic exchanges between CAHG groups, we took the re-phased CAHG ancestry components output from Gnomix and used refinedIBD[86] to identify shared IBD blocks between each pair of individuals >1 cM. This threshold allows the identification of individuals that shared common ancestors within the last 2,500 years while ensuring a high-enough power to detect IBD blocks and an extremely low false positive rate[50,54–57]. A key date for this study is 2500 BP because this period is thought to represent the initial north–south migration of Bantu-speaking communities across the Equator following a fragmentation of rainforest that would have created corridors for human migration[87–89] (but see ref. 64 for putative evidence of an earlier crossing of the rainforest by Bantu speakers). We then merged IBD blocks within a 0.6 cM gap and allowed only one inconsistent genotype between the gap and block regions using the program merge-ibd-segments from Beagle utilities[90]. Finally, we selected only those IBD blocks shared by individuals belonging to two different populations and masked them in their corresponding haplotypes on the already masked file (see Supplementary Text 1 for further details into IBD analyses).

### Ancestry-specific genotype frequency matrix

Following ref. 82 for each haplotype in a given diploid individual, masking was done for SNPs located in non-CAHG chromosomal segments. The individual's two haplotypes were averaged, creating a genotype frequency vector with (0, 0.5 or 1) average alternate allele count. Sites at which an individual had no CAHG ancestry on either haplotype are marked as missing. Matrix completion was used on the $N$ individuals × $p$-genotyped markers SNP matrix following ref. 82.

### Ancestry-specific MDS

As described in ref. 82, we performed MDS on the ancestry-specific genotype matrix described above. For a distance metric, we used the average number of pairwise differences, which varies directly with genetic drift[91].

### Ancestry-specific population statistics

Following ref. 82, the two population statistics described below were computed on population variant frequency ($f$) vectors created by computing, for each site,

$$f_i = \frac{a_i}{n_i}$$

where $a_i$ is the minor allele count in population $i$ considering only CAHG chromosomal segments, and $n_i$ is the total count of CAHG minor and major alleles in $i$. Any sites located in one or fewer CAHG segment were removed from the dataset for all populations, so $n_i > 1 \forall i$. This filtering resulted in the loss of 9,738 SNPs from the total 431,867 SNPs, leaving 422,129 SNPs across all populations for computation of the following population allele frequency statistics.

We used $F_{ST}$ statistics to compute pairwise genetic distances between the cultural groups (Supplementary Figs. 7–9). Pairwise $F_{ST}$ is the proportion of the total genetic variance due to between-population differences and is a convenient measure because it does not depend on

the actual magnitude of the genetic variance. In other words, genetic markers that evolve slowly are expected to have the same $F_{ST}$ value as markers that evolve more rapidly because the total variance is decomposed into within-population and between-population components[62]. We used this measure because it can account for homozygosity within populations and population histories were comparable[21,76].

We use Hudson's estimator for $F_{ST}$:

$$F_{ST}^{Hudson} = \frac{(f_A - f_B)^2 - \frac{f_A(1-f_A)}{n_A-1} - \frac{f_B(1-f_B)}{n_B-1}}{n_A(1-n_B) + n_B(1-n_A)},$$

for a given SNP, where A and B are different populations. For more than one site, the numerator and denominator were averaged across all SNPs separately and then the division was performed[92].

### Environmental similarities

The similarity in the environments inhabited by each of the cultural groups was calculated by using differences in biome composition between the cultures' territories. The biome composition at each 0.5° grid cell in the map was obtained from a recently published global vegetation model[93]. Two distance matrices were built: one based on Gower dissimilarities in the percentage of the culture's territory occupied by each of the biomes included in the dataset from ref. 93 and the other based on presence or absence of each biome type.

### Measuring cultural distances

To compute between-culture similarities in cultural repertoires we used Jaccard distances, equivalent to the ratio between the sum of the number of traits present in one culture but not in another and the sum of the number of traits that are present in one or both the cultures. We did this for musical instruments and subsistence tools separately. The resulting numerical index is bounded between 0 (identical presences in the two sites) and 1 (complete absence of shared traits) and is extremely useful for this type of data as it does not consider negative matches (that is, shared absences) that are common in sparse matrices with many absences[45,46,94].

### Assessing the drivers of cultural distances

We estimated the relative effects of geographic, biome and ancestry-specific genetic distances on similarities in musical instrument and subsistence toolkit repertoires using Mantel tests[95], which assess correlations between dissimilarity matrices using 1,000 permutations to obtain $P$ values that were then adjusted using the Benjamini–Hochberg procedure[45]. We then used multiple matrix regressions to test for the effect of each variable while controlling for the impact of a third matrix, which allows investigating causal relationships between distance matrices rather than the paired vectors themselves[45,96], also using 1,000 permutations to obtain $P$ values. Mantel tests and multiple matrix regressions are routinely used in population studies, ecology and increasingly in archaeological and anthropological studies aiming precisely at assessing the drivers of the structure of cultural diversity[45,46,94,97,98].

To further assess the robustness of the relationship between geographic, genetic and cultural similarities we analysed the CADM[99] using the ape R package[100]. It has been shown that CADM has the correct rate of type I error and good power when applied to independently generated distance matrices. For each genetic dataset (full genetic dataset, CAHG ancestry segments, CAHG ancestry segments minus IBD segments and Bantu ancestry segments only), we first used the global test of significance, CADM.global, to assess the congruence of the genetic distance matrix based on such a genetic dataset, with the matrices of geographical distance, ecological distance (that is, distance in biome composition) and cultural distance based on either musical instrument or subsistence tool repertoires. For the global test of significance, all distance matrices are permuted at random,

independently of one another. The null hypothesis for this test is the incongruence of all matrices. The alternative hypothesis is that at least two matrices are congruent, with similar rankings of the distances (Supplementary Tables 6, 7). We then performed a posteriori tests with the function CADM.post to identify groups of congruent matrices. In these tests, the null hypothesis is the incongruence of the matrix subjected to the test, with respect to all other matrices included. The alternative hypothesis is that this matrix is congruent with at least one other matrix in the set, having similar rankings of the distances. To preserve a correct or error rate, the probabilities are adjusted for multiple testing using the procedure as it is generally recommended for non-independent tests[101].

Finally, to visualize the structure of cultural and genetic diversity between groups while controlling for multicollinearity, we performed a PCoA on the distance matrices of pairwise Jaccard distances for musical instruments and subsistence tools and of pairwise $F_{ST}$ (Supplementary Figs. 2 and 3). Similar to a principal component analysis, a PCoA produces a set of orthogonal axes whose importance is measured by eigenvalues. However, in contrast to the principal component analysis, non-Euclidean distance matrices can be used[62]. We then visualized the PCoAs using the NeighborNet algorithm in phangorn[102].

### Assessing the sharing of terms to design musical instruments and subsistence tools

Words were counted as shared between two populations speaking different languages when they used words designating the same object that are sufficiently similar to suggest borrowing and common origin, as opposed to accidental lookalikes. To judge this, we used the Glottocodes of each of the CAHG languages to collect a set of grammars, word lists and other language descriptions to ascertain whether observed differences between similar words can be explained by language-specific patterns in morphology (for example, an added affix might make forms look more different than they are) or by differences in superficial conventions in the orthography used for transcription (for example, whether low tone is transcribed or whether prenasalized fricatives are written *mf*, *mv*, *nf* or *nv*). See Supplementary Dataset 2, with notes on the rationale behind our judgements and bibliographical references, and Supplementary Text 2 for a more detailed explanation of the methodology.

After compiling the final list of shared musical instrument and subsistence tool terminology between CAHG groups, we also noted whether the objects those terms depicted were unique to CAHGs (as opposed to present in other African non-hunter-gatherer groups) and whether the terms themselves were unique to CAHGs (see Supplementary Dataset 2 for notes). We also noted when the relationships between terms recorded in different languages were likely but there was insufficient evidence to conclusively back up the claim of relatedness. We then created two datasets of shared specialized vocabulary: one (broader dataset) including all shared terms between CAHG groups, and a restricted dataset including only those terms for which we had conclusive evidence of their relatedness and that were unique to CAHGs. We believe both datasets are important as CAHGs are known in Africa for their extensive music and musical instrument repertoires. Hence, many Bantu (and other farmer) populations rely on these instruments to sing and dance during their own ceremonies and rituals. This often results in ethnographers reporting that farmers use the hunter-gatherer word for particular instruments because the instrument comes from the hunter-gatherers and is often only used in contexts in which hunter-gatherers are involved[42,43].

### Calculating linguistic distances between populations

Although all CAHGs are thought to speak languages borrowed from neighbouring agriculturalist populations following large farming expansions, in some instances they are considered to have diverged enough to constitute distinct languages[16] (Supplementary Tables 2 and 3). Nonetheless, several CAHGs are thought to speak dialects of languages spoken by neighbouring farming groups. To calculate an estimate of linguistic phylogenetic distance between pairs of CAHG languages, we obtained the ISO 639-3 code of either the CAHG language itself or the farmer language from which the CAHG dialect was most closely related (Supplementary Table 3). In the latter case, we confirmed the match between the classification of farmer language and CAHG dialect using ref. 17.

Next we estimated phylogenetic distances between each pair of languages using three methods. First, we used machine-learning techniques on the word lists of the ASJP database[60,61]. The ASJP contains lists of the 40 most stable items from each language from the 100-item Swadesh list[103] and is thought to best reflect language relatedness[61,104]. We measured linguistic distances based on alignments weighted by sound correspondence probabilities estimated by PMI. PMI-weighted distances correspond more closely to distances in recognized phylogenies than other available estimates of lexical distances[61,62,105]. For the Babongo, the only linguistically heterogeneous population in our sample, we averaged pairwise distances based on each of its languages.

Second, we calculated phylogenetic distances using the global language trees from Glottolog v.4.8 (ref. 63). To do this, we merged the Central Sudanic and Atlantic–Congo subtrees and counted the number of internal nodes between each pair of CAHG languages. Third, we calculated patristic distances between each pair of languages using the augmented phylogenetic tree from the latest published phylogeny of Bantu languages by ref. 64, matching languages by their Glottocodes. For the three languages that were not part of the Bantu family, we took the maximum tree depth as the distance between each of those languages and every other language in the sample.

### Assessing the drivers of shared vocabulary items across CAHG groups

To determine whether shared words between groups were the product of a deep shared ancestry as CAHGs or instead of the borrowing of similar terms from closely related farming languages, we fitted zero-inflated models with a Poisson link function for the count model and a binomial link function for the binary part. Our choice of model formulation is due to the nature of the dataset containing a large number of zeros (pairs not sharing words) for structural and sampling reasons[106]. In the models, we used either pairwise PMI distances, pairwise Glottolog distances or patristic phylogenetic distances, and either pairwise genetic distance based exclusively on CAHG genomic segments or spatial distance between groups to predict their number of shared musical instrument words (Supplementary Tables 14–19). All predictor variables were standardized for comparability of coefficients. Given that genetic and spatial distances had a correlation coefficient of 0.760 ($P < 0.001$), we could not include both predictors in the same models as that could lead to a spurious reversal of estimates from one of the correlated predictors[107]. We could not fit similar models of subsistence tool terminology between groups because cases of shared terminologies were extremely limited.

Lastly, we determined the significance of cultural domain in word sharing by performing a paired Wilcoxon signed-rank test comparing the mean number of shared words between cultural dyads across cultural domains (that is, musical instruments versus subsistence tools).

### Inferring processes of admixture in genetic and cultural data

To assess the relationship between geographic, cultural and genetic distances between populations, we performed three independent SpaceMix[53] analyses aimed at retrieving the ten genetic and cultural pseudo-spatial positions of the CAHG groups in our sample with available genetic data and musical instrument and subsistence tool repertoires. For the genetic run, we used our masked data and the allele.

frequency option of SpaceMix, with the sample_means option specifying for each of the 422,128 loci remaining in our dataset after masking, the average number of individuals in each population carrying a CAHG allele for that locus. For the musical instrument and subsistence tool runs, we used the count/total option of SpaceMix by generating a pseudo-genetic file in which each population was represented as a single diploid individual and the presence of a given instrument or tool was registered as a homozygous trait[108]. The geno–geographical and cultural–geographical coordinates generated were compared with the actual geographical coordinates visually and numerically.

In addition, we used SpaceMix runs on genetic data to assess evidence for recent gene flow between CAHG groups following agriculturalist expansions by computing and visualizing sources and direction of recent admixture between populations[109]. For each population, admixture proportions were obtained by calculating the proportion of alleles estimated to come from that population's inferred admixture source location (Supplementary Table 9 and ref. 109) as opposed to the population geo–genetic location. We then selected those populations with inferred admixture proportions >1% from the Markov chain Monte Carlo run with the highest posterior probability, which we plotted in Supplementary Fig. 13.

### Reporting summary

Further information on research design is available in the Nature Portfolio Reporting Summary linked to this article.

## Data availability

Processed genetic data and raw cultural and linguistic data required to reproduce all the results reported in the article and Supplementary Information are available in the following GitHub repository: https://github.com/ceciliapad/GenCultEvo. Raw genome-wide SNP data were obtained from the datasets reported in refs. 24,75–77 and are available in the public servers EGA (accession numbers EGAS00001002078 and EGAC00001000139) and dbGaP (accession numbers phs000449.v2.p1 and phs001780.v1.p1). Language phylogenies were obtained from ASJP[60,61], Glottolog v.4.8 (ref. 63) and ref. 64.

## Code availability

The code required to reproduce all the results reported in the article and Supplementary Information are available in the following GitHub repository: https://github.com/ceciliapad/GenCultEvo.

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

## Acknowledgements

We thank F. Marandola, S. Fürniss, R. Jadinon, N. Chaudhury and E. Cornelissen for their help curating the cultural corpus, and E. Patin for help with assembling the genetic dataset. We also thank M. Sánchez Villagra and G. Aguirre for helpful methodological discussions. We thank R. Schlaepfer for help with the article figures. We thank the University of Zurich for financially supporting A.B.M., L.V. and C.P.-I. We are grateful to Fundación La Caixa and Forschungskredit Candoc for the funding of this study (grants LCF/BQ/EU19/11710043 and FK-19-083 awarded to C.P.-I.). The funders had no role in study design, data collection and analysis, decision to publish or preparation of the manuscript.

## Author contributions

C.P.-I., L.V. and A.B.M. conceptualized the study. C.P.-I. compiled the data. C.P.-I., J.B.-P., A.G.I., B.P., B.B., A.M., L.V. and A.B.M. developed the methodology. C.P.-I. analysed the data. C.P.-I., L.V. and A.B.M. wrote the article. All authors reviewed the article, provided comments and approved the final draft.

## Funding

## Competing interests

The authors declare no competing interests.

## Additional information

**Correspondence and requests for materials** should be addressed to Cecilia Padilla-Iglesias or Andrea Bamberg Migliano.

Andrea Migliano

# Reporting Summary

## Statistics

For all statistical analyses, confirm that the following items are present in the figure legend, table legend, main text, or Methods section.

| n/a | Confirmed | |
|---|---|---|
| ☐ | ☒ | The exact sample size (*n*) for each experimental group/condition, given as a discrete number and unit of measurement |
| ☐ | ☒ | A statement on whether measurements were taken from distinct samples or whether the same sample was measured repeatedly |
| ☐ | ☒ | The statistical test(s) used AND whether they are one- or two-sided<br>*Only common tests should be described solely by name; describe more complex techniques in the Methods section.* |
| ☐ | ☒ | A description of all covariates tested |
| ☐ | ☒ | A description of any assumptions or corrections, such as tests of normality and adjustment for multiple comparisons |
| ☐ | ☒ | A full description of the statistical parameters including central tendency (e.g. means) or other basic estimates (e.g. regression coefficient) AND variation (e.g. standard deviation) or associated estimates of uncertainty (e.g. confidence intervals) |
| ☐ | ☒ | For null hypothesis testing, the test statistic (e.g. $F$, $t$, $r$) with confidence intervals, effect sizes, degrees of freedom and $P$ value noted<br>*Give P values as exact values whenever suitable.* |
| ☒ | ☐ | For Bayesian analysis, information on the choice of priors and Markov chain Monte Carlo settings |
| ☒ | ☐ | For hierarchical and complex designs, identification of the appropriate level for tests and full reporting of outcomes |
| ☒ | ☐ | Estimates of effect sizes (e.g. Cohen's *d*, Pearson's *r*), indicating how they were calculated |

*Our web collection on statistics for biologists contains articles on many of the points above.*

## Software and code

Policy information about availability of computer code

| Data collection | *Provide a description of all commercial, open source and custom code used to collect the data in this study, specifying the version used OR state that no software was used.* |
|---|---|
| Data analysis | All data and code required to reproduce the results in the manuscript and supplementary information is available in the following GitHub repository: https://github.com/ceciliapad/GenCultEvo. We used the following software:<br><br>- Gnomix (https://github.com/AI-sandbox/gnomix)<br>- PLINK (PLINK v1.90b6.12)<br>- R (R version 4.3.1)<br>- KING (https://www.kingrelatedness.com)<br>- Spacemix (https://github.com/gbradburd/SpaceMix)<br>- ADMIXTURE (ADMIXTURE/1.3.0)<br>- Beagle 5.3<br>- RefinedIBD(https://faculty.washington.edu/browning/refined-ibd.html)<br>- ape R package (version 5.7)<br>- phangorn R package (version 2.10)<br><br>The remaining packages are specified not in the methods but in the scripts included in the Github repository. |

For manuscripts utilizing custom algorithms or software that are central to the research but not yet described in published literature, software must be made available to editors and reviewers. We strongly encourage code deposition in a community repository (e.g. GitHub). See the Nature Portfolio guidelines for submitting code & software for further information.

## Data

Policy information about availability of data

All manuscripts must include a data availability statement. This statement should provide the following information, where applicable:

- Accession codes, unique identifiers, or web links for publicly available datasets
- A description of any restrictions on data availability
- For clinical datasets or third party data, please ensure that the statement adheres to our policy

> Processed genetic data, and raw cultural and linguistic data required to reproduce all the results reported in the manuscript and supplementary are available in the following Github repository: https://github.com/ceciliapad/GenCultEvo. Raw genome-wide SNP data is available in public servers EGA (Accession number EGAS00001002078 and EGAC00001000139) and dbGaP (Accession numbers phs000449.v2.p1 and phs001780.v1.p1).
> Language phylogenies were obtained from the ASJP database (https://asjp.clld.org), Koile et al. (2022) and Glottolog v.4.8 (https://glottolog.org)

## Research involving human participants, their data, or biological material

Policy information about studies with human participants or human data. See also policy information about sex, gender (identity/presentation), and sexual orientation and race, ethnicity and racism.

| | |
|---|---|
| Reporting on sex and gender | No subjects were involved. |
| Reporting on race, ethnicity, or other socially relevant groupings | No subjects were involved but cultural information on Central African hunter-gatherer populations were obtained from museum collections, ethnographers and primary literature. Population named reflect their recognised ethnicities and primary literature. |
| Population characteristics | We compiled published cultural, genetic and location data from each population and museum data on cultural artifacts. |
| Recruitment | No subjects |
| Ethics oversight | Research did not require human subjects or animals and no ethics oversight was required, but a data access committee gave ethics approval for our use of the above-mentioned genetic datasets for the purpose of this project. |

Note that full information on the approval of the study protocol must also be provided in the manuscript.

# Field-specific reporting

Please select the one below that is the best fit for your research. If you are not sure, read the appropriate sections before making your selection.

☐ Life sciences  ☐ Behavioural & social sciences  ☒ Ecological, evolutionary & environmental sciences

For a reference copy of the document with all sections, see nature.com/documents/nr-reporting-summary-flat.pdf

# Ecological, evolutionary & environmental sciences study design

All studies must disclose on these points even when the disclosure is negative.

| | |
|---|---|
| Study description | This study presents an analysis of cultural, genetic and linguistic data from Central African hunter-gatherer populations. |
| Research sample | Published data on words, genetic sequences and museum data on cultural artifacts from Central African hunter-gatherer populations. |
| Sampling strategy | We collected all data available on cultural artifacts and vocabulary items. |
| Data collection | Primary literature and museum visits |
| Timing and spatial scale | A few months of visits to libraries and museums. Spatial scale is the entire Congo Basin. |
| Data exclusions | No exclusion |
| Reproducibility | All data and code are available |
| Randomization | Not applicable |
| Blinding | Not applicable |

Did the study involve field work?  ☐ Yes  ☒ No

# Reporting for specific materials, systems and methods

We require information from authors about some types of materials, experimental systems and methods used in many studies. Here, indicate whether each material, system or method listed is relevant to your study. If you are not sure if a list item applies to your research, read the appropriate section before selecting a response.

## Materials & experimental systems

| n/a | Involved in the study |
|-----|----------------------|
| ☒ | Antibodies |
| ☒ | Eukaryotic cell lines |
| ☒ | Palaeontology and archaeology |
| ☒ | Animals and other organisms |
| ☒ | Clinical data |
| ☒ | Dual use research of concern |
| ☒ | Plants |

## Methods

| n/a | Involved in the study |
|-----|----------------------|
| ☒ | ChIP-seq |
| ☒ | Flow cytometry |
| ☒ | MRI-based neuroimaging |

## Plants

Seed stocks
*Report on the source of all seed stocks or other plant material used. If applicable, state the seed stock centre and catalogue number. If plant specimens were collected from the field, describe the collection location, date and sampling procedures.*

Novel plant genotypes
*Describe the methods by which all novel plant genotypes were produced. This includes those generated by transgenic approaches, gene editing, chemical/radiation-based mutagenesis and hybridization. For transgenic lines, describe the transformation method, the number of independent lines analyzed and the generation upon which experiments were performed. For gene-edited lines, describe the editor used, the endogenous sequence targeted for editing, the targeting guide RNA sequence (if applicable) and how the editor was applied.*

Authentication
*Describe any authentication procedures for each seed stock used or novel genotype generated. Describe any experiments used to assess the effect of a mutation and, where applicable, how potential secondary effects (e.g. second site T-DNA insertions, mosiacism, off-target gene editing) were examined.*

