## [Peer Review File · Nature Human Behaviour]

Peer Review Information

Journal: Nature Human Behaviour

Manuscript Title: Deep history of cultural and linguistic evolution among Central African hunter-gatherers

Corresponding author name(s): Cecilia Padilla-Iglesias and Andrea Bamberg Migliano

Reviewer Comments & Decisions:

Decision Letter, initial version:
--

12th July 2023

Dear Professor Migliano,

Thank you once again for your manuscript, entitled "Deep history of cultural and linguistic evolution among Central African hunter-gatherers", and for your patience during the peer review process.

Your Article has now been evaluated by 4 referees (as you know, a fifth reviewer initially agreed to review but has not been able to return their comments; if we receive a report from them in due course we will send this to you). You will see from their comments copied below that, although they find your work of potential interest, they have raised quite substantial concerns. In light of these comments, we cannot accept the manuscript for publication, but would be interested in considering a revised version if you are willing and able to fully address reviewer and editorial concerns.

We hope you will find the referees' comments useful as you decide how to proceed. If you wish to submit a substantially revised manuscript, please bear in mind that we will be reluctant to approach the referees again in the absence of major revisions. We are committed to providing a fair and constructive peer-review process. Do not hesitate to contact us if there are specific requests from the reviewers that you believe are technically impossible or unlikely to yield a meaningful outcome.

To guide the scope of the revisions, the editors discuss the referee reports in detail within the team, including with the chief editor, with a view to (1) identifying key priorities that should be addressed in revision and (2) overruling referee requests that are deemed beyond the scope of the current study. We hope that you will find the prioritised set of referee points to be useful when revising your study. Please do not hesitate to get in touch if you would like to discuss these issues further.

1) Please redo your analyses of linguistic distance using the method (patristic distance) suggested by Reviewer 2.

2) Please fully address all concerns raised by reviewers about the justification for your methodological choices and the details of your datasets, so that the methods you use can be evaluated in full.

3) Please add the plots requested by Reviewer 4.

4) Please make all data and code available for reviewers to check.

5) We appreciate Reviewer 1's suggestion to remove the genetics component of the paper altogether; however in light of all reviewer comments we would instead ask that you clarify your assumptions in full, transparently acknowledge the limitations of your methods, and make the description of (and rationale for) your genetic analyses more accessible to an interdisciplinary readership.

6) Reviewer 1 also notes that the manuscript would benefit from a more in depth and qualitative discussion of the ethnographic data used, and editorially we support this suggestion.

If you wish to submit a suitably revised manuscript, we would hope to receive it within 4 months. I would be grateful if you could contact us as soon as possible if you foresee difficulties with meeting this target resubmission date.

- Include a "Response to the editors and reviewers" document detailing, point-by-point, how you addressed each editor and referee comment. If no action was taken to address a point, you must provide a compelling argument. When formatting this document, please respond to each reviewer comment individually, including the full text of the reviewer comment verbatim followed by your response to the individual point. This response will be used by the editors to evaluate your revision and sent back to the reviewers along with the revised manuscript.
- Highlight all changes made to your manuscript or provide us with a version that tracks changes.

[REDACTED]

Thank you for the opportunity to review your work. Please do not hesitate to contact me if you have any questions or would like to discuss the required revisions further.

Sincerely,

[REDACTED]

Senior Editor
Nature Human Behaviour

Reviewer expertise:

Reviewer #1: African archaeology & hunter gatherers

Reviewer #2: linguistics, cultural evolution, Bantu languages, phylogenetics

Reviewer #3: hunter gatherer archaeology; material culture; cultural evolution & transmission

Reviewer #4: genetics of African populations

REVIEWER COMMENTS:

Reviewer #1:

Remarks to the Author:

The article 'Deep history of cultural and linguistic evolution among Central African hunter-gatherers' evaluates genetic, linguistic and material culture characteristics of 10-11 CAHG populations to understand the relationship between technology transfer between populations and the retention or adoption of different words in relation to geography, kinship and ecology. New methods for understanding relationships between genomic patterns and material culture (instruments) are proposed. The authors conclude that material culture and language patterns 'reflect long evolutionary histories'.

Back in 1977, Ian Hodder's 'The Distribution of Material Culture Items in the Baringo District, Western Kenya' (Man 12: 239-269) article about exchange of material culture in western Baringo really needs to be included at least tacitly in this discussion since it essentially arrived at the same conclusion albeit in quite a different way. The relationship between certain aspects of material culture, geography and language is a long-running discussion and it is well known that geography is a better predictor for material culture occurrences more so than language or kinship, but there are certain culturally specific attributes that are retained internally and others that are disseminated. Overall, there is nothing surprising about the results the authors present here, although some of the nuances of the relationship between language, geography and material culture are interesting (I do not buy into the genetic arguments made here at all; see below). I think that, if reframed a bit to incorporate modern anthropological thought about relationships between migration and material culture patterning, this paper could be publishable, although the changes would need to be rather significant.

First, the organisation of this manuscript is a bit odd. The Introduction gives away the whole pony before we really have a chance to evaluate the results or have a discussion. While it is common to have a sentence or two summary of the 'take-away' punchline at the end of an Introduction (i.e., thesis statement), I felt like I had already read everything I needed to know by Line 89. The rest is

just supporting information. A lot of what is in the Introduction should be removed so that the data can be presented first and then the conclusions are made.

The figure captions are a bit wanton of information (Fig 1 also includes a grammatical mistake). Figures should be essentially self-explanatory with the captions, but the caption leaves too much to the imagination to be independently informative. See my specific critique of Fig. 1 below. Fig. 3 is good (captionwise, although the underlying assumption in the method is perhaps not). The other figures are not great or have missing bits of information critical to impart meaning.

I am not a geneticist, but I have a decent background in statistics and I am not convinced that subtracting out genomic segments to correlate ancestry vs. different types of musical instruments is really a rigorous test of the hypothesis (L160-167). L179-180 reads more like a by-line from 'Guns, Germs and Steel' than any modern anthropological perspective on material culture exchange. But, beyond that, I am not sure how one can exclude certain aspects of a group's genome and then correlate it specifically with instruments, either in the present or past. I read this section carefully several times and it makes no sense to me. Either the method is not explained well or I am missing some critical piece of knowledge to have this make sense to me, but, minimally I think some of the underlying assumptions need to be clarified. Are you looking at older instruments (i.e., from the archaeological record—from your Methods section it seems that you are not, but if not, then how can you connect 'old genomes' to specific instruments or names of instruments)? Assuming you can do black box subtractive math of genomes to identify sequential 'ur-populations' in the genome, how to you attribute specific instrument occurrences with those different segments given the known demographic complexity of Central Africa? To me, this is a step too far. In my opinion, this section of the paper should be dropped altogether because it makes no sense—at least to me.

In sum, the paper uses a lot of statistical methods, many of which are rather complex, that compound upon each other to arrive at a rather simple conclusion. Particularly by the time the genetics arguments are laid out, I felt a bit lost in the forest of methods that seemed to teeter on assumptions that could not be fully justified. I am familiar with all of the methods, at least in principle, that were used in the paper and actually teach statistics modules to archaeology Masters' students at a university, so it is not that I am averse to complex math or the use of statistics. It is just that there needs to be a balance of statistics along with good old-fashioned ethnography, standing on decades of research of anthropologists going into the field and interviewing people about what, when, where and why people make economic decisions related to material culture. In my mind, it seems that this paper has arrived at the most parsimonious conclusion to the question they set out to answer, but the way they get there is not clear and is a bit of a house of statistical cards that needs more solid footing.

Minor matters:

Fig 1: Does not have a scale. Nor is there a legend to indicate what the different shades of green represent. The use of modern national borders should be clarified (although it isn't particularly pertinent to anything in the paper), and if included, you should have country names labelled.

Fig. 4: I know that the ethnic group names were already presented in Fig. 1, but in the discussion below Fig. 4, I had to go back and forth between Fig. 1 and 4 to associate geography with the groups being mentioned. This is not an effective presentation style.

L105: Why 'acculturation', specifically? You imply a dominance hierarchy in which farmers enculturate

foragers, but this is not always the way it works. There is often a two-way exchange of material culture (see Hodder's 1977 article) that is not based on 'acculturation'. Please choose another word here or expend words to explain why it is 'acculturation' as opposed to other mechanisms of social exchange.

L148, 208, 237, 471: No comma needed.

L253: Oral traditions ≠ legends.

L256: 'an' should be 'a'; and, see again, Hodder (1977) among many sources that demonstrate that linguistic connections do not map clearly over genetic affiliations or material culture. We don't know when these words originated or what the context is for their shared phonemal linkages. We only know that they exist across space irrespective of language similarities. The implied connection between language (often used as a proxy for kinship since languages are socially learned) and material culture hearkens back to some old thoughts about migration and material culture popular in the 1970s through 1990s, which I advise the authors to contextualise better than they have done here.

- See also <https://doi.org/10.1080/0067270X.2011.629525>
- See also <https://doi.org/10.1111/j.1467-9655.2010.01674.x>
- See also <https://doi.org/10.1038/s41598-018-31123-z>

Reviewer #2:

Remarks to the Author:

This article compares the evolution of different cultural traits, namely musical instruments and foraging tools, as well as the lexical terms for both, along the deep and shallow history of 10 Central African hunter-gatherer (CAHG) groups. This evolution is compared with the genetic and linguistic history of these groups, as well as its ecological and geographical situations. The results of this study show that both the musical instruments and their names present in these CAHG groups in current times are the product of their deep history, predating the Bantu expansion and language shift of these groups, as well as the interaction between different groups. On the other hand, foraging tools, as well as their names, would have been independently adapted to the different ecological regions. This is an interdisciplinary study, including genetics, anthropology, linguistics and ecology, with a focus on genetics, where a novel method is implemented for studying different layers of history.

Although the results are interesting, both for specialized scholars and for more broader audience, I see flaws in the presentation of the methods and results, for which I would recommend major revisions before the manuscript is accepted. Below, I list 4 major comments, as well as several minor comments to be addressed. I must say that my background is in quantitative analyses and linguistics, for which my comments on the genetic analyses are not very specific, barely educated guesses.

Major comments

1. Code availability

The Data and code availability section states that "All the data and code required to reproduce all the

results reported in the manuscript and supplementary information will be made available upon acceptance of the paper in the following Github repository: <https://github.com/ceciliapad/GenCultEvo>, which is private at the time. Therefore, I wasn't able to review the code, only the Dataset S1. Please make it available to allow for replicability.

As for the Dataset S1, I detected some inconsistencies, as having up to four different terms for one single trait: (arrow, Arrow, arrows, Arrows). These issues might have been solved in the code by re-coding all these forms into the same value, but I cannot evaluate that without seeing the code.

2. Language classification and linguistic distances

I have several comments on this section, regarding the different language classifications presented, the method used for defining the linguistic distances, and finally the cognate judgment used to decide whether two lexemes coincide across languages for a given concept.

2a. Different language classifications (Tables S12-13)

I don't understand the rationale for including two different classifications of the languages involved (one of them including the speculative Nigil-Kordofanian macrofamily), none of which will be ultimately used for calculating linguistic distances – distance between wordlists is used instead, see next point. Also, glottocodes, one of the 2 standard labels together with ISO 639-3, are not reported (neither the Glottolog classification mentioned), and the labels used for denoting languages in the first Table S15, where the linguistic distances are reported, are different from any of the classifications mentioned, making the match between them hard to follow.

I tried to link the languages in Tables S12, S13, and S15 (first) but wasn't able to do all the matches. For example, I don't understand if the languages for Mbendjele and Aka are supposed to be the same or not (if there is no typo, the Mbendjele speak the language with ISO-code mgd, meaning Moru (Central Sudanic), but I imagine it meant ngd instead, reducing the total number of languages to 13 instead of 14). Secondly, the ISO-code kin corresponds to Kinyarwanda. Was this meant to refer to Twa (Glottocode twaa1238, variety of Ljenje, Glottocode lenj1248, ISO-code leh)? Finally, I don't see a match for the languages NZEBI and SANGO in Table S15. The first seems to refer to the glottocode nzeb1234, which is a branch containing 7 languages according to Glottolog, and Sango has glottocode sang1328, ISO-code sag, not present in Tables S12-13. Please specify the correct matches, so this is clear for the reader (consider the table below just as an attempt: the authors certainly know better than me what exact languages they mean and will be able to produce a much better matching).

Comparison of classifications:

Comparison of classifications:

CAHG group	Classif S12	Classif S13	ASJP	ISO	Glottocode	Grollemund et al. 2015	Koile et al 2022 (aug)
Baka	Gbanzili-Sere	Baka	BAKA	bkc	baka1272	NO	NO (Ubangi)
Bedzan	Tikar	Tikar	TIKAR	tik	tika1246	NO	NO (Northern Bantoid)
Bakola	A80	Gyele	GYELE	gyi	gyel1242	NO	YES
Bakoya	B20	Koya	KOYA	nra	ngom1270	YES	YES
Babongo	B30	Kaningi	KANINGI	kzo	kani1279	YES	YES
Babongo	B60	Teke	TEKE	tek	ibal1241	YES	YES
Babongo	B70	Tsogho	TSOGO	tsv	tsog1243	YES	YES
Mbendjele	C10	Ngando	NGANDO	mgd	ngan1304?	YES	YES
Aka	C10	Ngando	NZEBI OR SANGO?	ngd	ngan1304	YES	YES
Batwa (West)	C60	Bushong	BUSHONG	buf	bush1247	YES	YES
Batwa (East)	J11	Twa	TWA	kin	kiny1244	NO	YES
Batwa (East)	J60	---	NZEBI OR SANGO?	---	?	?	?
Sua	D30	Bila	BILA	bil	bile1244	YES	YES
Efe	Mangbetu-Efe	Efe	EFE	efe	efee1239	NO	NO (Central Sudanic)

2.b Linguistic distances

I disagree with the method used here to calculate linguistic distances (Lines 227-246 and 552-560). Although proven to align with known phylogenetic classifications to a reasonable extent in Jäger 20018 (ref 46), this method to measure linguistic distances is non-standard, and has several caveats.

It is based only on the distance between wordlists, so it doesn't necessarily correlate with phylogenetic distances (which is what I assume is the authors' intention). In creating a phylogeny, not only pairwise-lexical distances are measured, but also different possible evolutionary trajectories are considered, and archaeological data can be used for calibrating the dates. If the matches in the table above are correct, EFE should be by far the language more distant to all the rest (non-Atlantic-Congo), followed by BAKA (Ubangian), TIKAR (Northern Bantoid), and then the other 10-11 languages, closer to each other than to any of the former three. However, from Table S15 we can see that e.g. NANINGI is closer to EFE than to BAKA, and almost as far to TIKAR and even GYELE (both Bantu languages) as to EFE. The same is true for NGANDO, TEKE, and TWA, which are closer to EFE than to SANGO. Also, KOYA is closer to BAKA than to SANGO, TIKAR, and BILA, etc.

Since all (but one) of these languages belong to the Atlantic-Congo macrofamily, the ideal way to measure linguistic distances here would be by using the patristic distance (time elapsed since the most recent ancestor for the 2 languages) from a known phylogeny. The phylogeny by Koile et al. 2022 (R1) (please add the reference, see below), in its augmented tree, includes all 10 Bantu

languages of the 13 languages in Table S13 (see table above), only missing Baka (Atlantic-Congo > Ubangi), Tikar (Atlantic-Congo > Northern Bantoid), and Efe (Central Sudanic). The phylogeny by Grollemund et al. 2015 (ref 66), includes 8 of these. The distances for the 3 missing languages can be approximated with a higher distance for the two Atlantic-Congo non-Bantu languages to all Bantu, and a further higher one for the Central Sudanic language.

A simpler option (although less precise) would be to part from an accepted classification of groupings such as Glottolog (R2), and take as distance the amount of nodes between a pair of languages (plus a fixed quantity for the case of Efe vs all Atlantic-Congo languages).

(R1) Koile, E., Greenhill, S. J., Blasi, D. E., Bouckaert, R., & Gray, R. D. (2022). Phylogeographic analysis of the Bantu language expansion supports a rainforest route. *Proceedings of the National Academy of Sciences*, 119(32), e2112853119.

(R2) Hammarström, Harald & Forkel, Robert & Haspelmath, Martin & Bank, Sebastian. 2022. *Glottolog 4.7*. Leipzig: Max Planck Institute for Evolutionary Anthropology. <https://doi.org/10.5281/zenodo.7398962> (Available online at <http://glottolog.org>, Accessed on 2023-05-04.)

2c. Cognates judgment and comparison with farmers' languages

How was the cognate judgment carried out? (Lines 200-218). Although it is mentioned later in the text that only exact matches were considered (Line 539: "...we only counted as "shared" words that were identical [after removing prefixes and suffixes]."), in Table S14 there are cases of different scripts (e.g. ṅòṃ / ngom) and presumably small variation (e.g. ruma / baruma, sànze / sanzi, likembe / kembu), while other items are considered as different cognate sets although they might be similar enough (e.g. how is ngomo / ngoma / ṅòṃ / ngomu / ngom different from ṅòṃbi / ngombu? I'm referring exclusively to the form here, not to the semantics, which seems to denote a different instrument). Was this checked by an expert in these languages?

Secondly, in Line 532: "Note that the same instrument may be designated by more than one term within the same cultural group due to within-group linguistic heterogeneity and the fact that the same instrument might be named differently depending on the occasion." What has been the criterion to select one form over another in these cases? Please specify.

Finally, in Lines 312-318: "We have also shown that although CAHGs have adopted languages from farmers, music related words were most likely inherited from ancestral populations together with the corresponding musical instruments. By contrast, the fact that very few tool-related words are shared between CAGHs may reflect either a convergent origin of the similar tools in groups with divergent languages (and hence distinct words for similar tools); or an ancient common ancestor to both subsistence tools and tool-related words, with tool (but not word) differentiation over time being limited by ecological constraints." In order to be sure about this, it would be important to compare these words with the ones for the same musical instruments (if present) by the farmer (Atlantic-Congo of Central Sudanic) speakers of these same or related languages.

3. Correlation analyses

Tables 1, S5, S6, S7, and S9 test similar hypotheses, namely correlation between the variables of interest: genetic diversity, geography, ecology, and cultural traits (musical instruments and subsistence tools), by using different statistical methods, namely Mantel test, CADM or Multiple matrix regression. I think it would be more clear if when the same comparison is performed with a different method, both results are summarized together in the same table or plot. Otherwise, it is not clear why three different methods are used to answer the same question, and in which cases they agree or disagree. This is explained to some extent in Lines 489-524, but it is still somehow confusing why so many analyses are attempted on the same data.

4. Pseudo-coordinates

I am not familiar with the transformation into pseudo-coordinates (Lines 137-142 and again Lines 258-260). This said, phrases like these are confusing to me (Line 258-260): "Moreover, a SpaceMix analysis showed no clear visual match between geographic distances and pseudo-distances estimated from subsistence tools (Fig. S6)." Is this analysis solely based on visual match of geographic distances with pseudo-distances retrieved from the dataset of interest? If this is the case, I don't see how Fig S6A for musical instruments shows a clear match, while Fig S6B for subsistence tools shows a clear mismatch. They don't look that different to me. In Materials and methods, the only explanation of the method is "The geno-geographic and cultural-geographic coordinates generated were compared with the actual geographic coordinates." Could you please specify how this comparison is quantified?

Minor comments

General

Please check the labels of the supplementary materials: Table S4 is non-existent, and the following tables' numeration is shifted (e.g. Tables S6 and S9 referred to as Table S5 and S8 in line 111, among many other cases). Also Table S15 is the name for two different tables, please correct. Also, please refer to the supplementary materials in order. It is hard to navigate them in the current order (e.g. the first supplementary figure referred is S1, then S10, S11,...).

Introduction

Line 41: "CAHGs" is used for Central African Hunter Gatherers, but it is not defined before in the main text (only in the abstract).

Lines 43-45: The abstract mentions 10 CAHG groups, but this sentence mentioned 11 groups, speaking 14 languages. I understand that there is at least some data (cultural) for 11 groups, of which genetic data is present only for 10, but this is not completely clear in this first paragraph, and it makes it look inconsistent with the abstract, please clarify. Also, the 14 languages are not addressed either in the figure or in the table mentioned, and it is unclear e.g. to understand which groups are multilingual. Tables S12-13 do mention languages: There are 14 rows (in Table S12), but the languages for Mbendjele and Aka seem to be the same, reducing the list to 13 languages, is this correct? Table S13 has 13 rows, but again Mbendjele and Aka seem to have the same language, and one of the languages of Batwa (East) is missing.

Line 59: please add one more relevant reference, on the use of musical instruments for studying the human history: (R3)

(R3) Aguirre-Fernández, G., Blasi, D., & Sánchez-Villagra, M. (2020). Panpipes as units of cultural analysis and dispersal. *Evolutionary Human Sciences*, 2, E17. doi:10.1017/ehs.2020.15

Line 61-63: Consistent spacing between magnitude and unit: "160 km" in line 61 but "2,000km" in line 63.

Line 73: groups referred to as Biaka and Mbuti here are referred to as Aka and Mbuit respectively in the caption of Figure 1. Please rename consistently.

Figure 1:

Map: The colors chosen for single and multiple genetic groups are shades of green, while the background map uses green for the steppe. In addition, international borders such as those inside the Baka group are marked with the same color and style as group borders (e.g. between Aka and Mbendjele). Both these design decisions make the map hard to read, and can be solved by e.g. replacing the background map for one with different colors and international border style (or no international border at all). These comments also apply to Figure 4 and S14.

Caption: Central African hunter-gatherer → gatherers. Also, check the names used for Aka and Mbuit (see above in Line 73).

Results

Line 103: Add relevant reference on the Bantu expansion and their interactions with CAHG: (R4)

(R4) Klieman, K. A. (2003). "The pygmies were our compass": Bantu and Batwa in the history of west central Africa, early times to c. 1900 CE. Greenwood.

Line 103 (and also Lines 275-283): "We calculated Jaccard distances between populations...". Please refer here to Materials and Methods → Measuring cultural distances. Also, if I understood it correctly, only attested cases were considered. Therefore, were NA's considered the same as absences? This distinction is crucial when drawing conclusions on a highly sparse set such as subsistence tools (Lines 275-283). Please clarify in Materials and Methods.

Line 106: NeighborNet algorithm, please add reference: (R5)

(R5) David Bryant, Vincent Moulton, Neighbor-Net: An Agglomerative Method for the Construction of Phylogenetic Networks, *Molecular Biology and Evolution*, Volume 21, Issue 2, February 2004, Pages 255–265, <https://doi.org/10.1093/molbev/msh018>

Figure 2, caption: "...PCoAs for based on..." → delete the "for"

Line 121: Is (Fig S3) correctly referenced?

Line 122 (also Line 135 and 254): see Methods → see Materials and Methods

Line 126: "We found a correlation..." → "We found a significant correlation..."

Line 128: Table S5 includes the Mantel test, while Table S6 includes the CADM analysis (please correct labels and add the missing one).

Figure 3: I don't understand the information conveyed by this figure (I must say here that I don't have a background in genetics, though). Is this merely an illustration of how the genetic sets look before and after removing the segments associated with each ancestry layer, or should it show any further information? Having 10 and 12 columns in each step, respectively, I imagine each of them is associated with one CAHG group? What do the heights stand for?

Line 173: "The approach implies various predictions" → "The approach can result in various different outcomes"

Line 202: "Hence, either CAHGs adapted words from farmer languages to name their instruments, or preserved the original names from their lost languages." These are not the only possible options. There could have been innovations for these lexical items in (some of) the CAHG languages either before or after the adoption of the farmers' neighboring languages.

Figure 4: Instrument 1 is called ikembe here, but referred in the rest of the text as in Table S14 as kembe.

Line 245: they → their

Discussion

Line 319: "We believe that extending our procedure may lead to the identification of other ancient shared linguistic terms in cultural domains known to precede the arrival of farming groups." What other cultural domains do you refer to?

Materials and methods

Line 340: data provided by any one source → data provided by every source

Line 345: it refers to N=1444 material cultural records, but the Supplementary Data S1 includes 1370 datapoints.

Line 356: and creating partial polygon objects → and partial polygon objects were created

Line 358: "Note that due to data availability, for the analyses concerning cultural repertoires, we did not distinguish between the Aka of the Central African Republic and the Mbendjele of the Republic of Congo, as genetically and culturally these groups are often considered a single cultural group". Does this also apply to Sua and Efe?

Line 396: between iteration → between iterations.

Line 408: "As recommended, we used 15% of the data for training set 1, 80% for training set 2 and 5% for validation..." As recommended by whom? Is this the standard practice in the field? Add

reference if relevant.

Lines 421-423: "2,500B.P. is a key date for the purpose of this study given that it is thought to represent the initial north-south migration of Bantu speaking communities across the Equator following a fragmentation of rainforest that would have created corridors for human migration." Please mention that recent studies have cast doubt on this, e.g. reference (R1).

(R1) Koile, E., Greenhill, S. J., Blasi, D. E., Bouckaert, R., & Gray, R. D. (2022). Phylogeographic analysis of the Bantu language expansion supports a rainforest route. *Proceedings of the National Academy of Sciences*, 119(32), e2112853119.

Line 458: decom-posed → decomposed

Line 474: Biome composition was obtained from reference 71, but this reference reconstructs the biomes for the last 120,000 years. At what time(s) was it obtained?

Line 476: Beyer (82) → Beyer (71)

Lines 506 and 512: CADM.global and CADM.post are not names of statistical methods, but rather the function names for a specific implementation of them. Please clarify the reference, possibly (R6)?

(R6) Paradis, E., & Schliep, K. (2019). ape 5.0: an environment for modern phylogenetics and evolutionary analyses in R. *Bioinformatics*, 35(3), 526-528.

Line 536: calleg → called

Lines 548-551: "To calculate an estimate of linguistic phylogenetic distance between pairs of CAHG languages, we obtained the ISO 639-3 code of either the CAHG language itself or the farmer language which the CAHG dialect was derived from." The last expression, implying that dialects "derive" from (larger, more prestigious) languages might sound harsh to the linguistic audience. I would rephrase it to something like: "...we obtained the ISO 639-3 code of either the CAHG language itself or the farmer language most closely related to it."

Supplementary materials

Fig S1-S2: Please improve the resolution of the text (it is hard to distinguish e.g. Baka vs Biaka).

Fig S2: In caption: (left) and (right) should be (top) and (bottom) respectively

Fig S3-5: It might be hard to distinguish the different colors for some readers. It would help to put the labels corresponding to a population near the corresponding dots. Also, at least in Fig S3, it would help to visually mark which groups correspond to NW, SW, and E (e.g. with an ellipse with dashed contour including the relevant groups in each region).

Fig S6: As mentioned above, it is not clear how these pseudo-coordinates should be interpreted. Maybe clarify this in the caption?

Fig S7-9: none of them are referred in the text

Fig S10 and S13: According to these figures, there are 10 groups with musical instruments but 11 with names for them, as well as 10 groups with subsistence tools but only 9 with names for them. Please clarify what this means (are there subsistence tools attested in one community but no name for them, and names for musical instruments in one community where the instrument is not attested?) Also, musical instruments correspond to the top plot in Fig S13 and to the bottom plot in Fig S10, which can be confusing.

Fig S12: how much of the variance is explained by each principal coordinate?

Fig S16: labels are in many cases hidden by the arrows, please jitter the former to increase readability.

Fig S17: Does this distribution imply that for a Masked $F_{st} = 0$ (identical populations) the predicted number of shared musical instruments is zero / very low, and the maximum is only at Masked $F_{st} = 0.01$?

Table S1: In the first paragraph it is stated that 14 languages are spoken by these 11 CAHG groups, and Fig 1 and this table are referred to, but no information of languages appears in any of them. Maybe add the language information to this table?. Also, what are the units for Area?

Table S4: missing according to the current numeration

Design: spaces between tables starting at Table S10-11 are shifted, leaving the captions and sometimes the first row on one page and the rest of the table in the other

Tables S12-13: see above (language classification).

Table S14: last column changes the order in which the populations are presented in different rows, e.g. "Aka, Baka" in one row and "Baka, Aka" in another. Maybe homogenize it with alphabetical order or another consistent order?

Table S15: duplicated name for 2 different tables

Table S15 (first one): It is not clear to which languages each of the abbreviations refer. These are neither the ones defined in Table S12 or in Table S13 (see table in Major comments).

Reviewer #3:

Remarks to the Author:

This innovative and important study by Padilla-Iglesias et al makes a significant contribution to understanding the drivers of cultural evolution in Central Africa. More generally it presents a case study illustrating the mechanisms of cultural and social evolution, as well as providing a methodological template that will inspire future studies. It is a perfect fit for Nature Human Behaviour and I would be delighted to see it published as soon as possible.

The 'new' data consists of a very significant inventory of nearly 1400 musical items and subsistence

tools. This has been carefully gathered from a range of sources. This is supplemented with the identification of shared lexical items between the groups under study. Genetic data have been gathered from material already publicly available. The between-group variation in these are compared with geography and environment.

The paper is very innovative in the way it compares genomic and cultural indicators of network distance as well as geography and linguistics. Statistical operations have been done appropriately, including correcting for multiple comparisons where appropriate.

Genomic analysis is very well explained and seems well executed with the masking of segments from more recent mixing irrelevant to what is being studied. This has led to the fascinating conclusion that musical instruments reflect a deep historical connectivity between groups that contrasts with the more geographically-determined adaptations seen in subsistence tools.

The paper is extremely well written and organised. Figures generally excellent, Figure 4 is a particularly good at conveying how shared material culture can be used to reconstruct historical connections between societies over considerable distances.

I have only a few suggestions for improvement:

There are a few miss-numbered references and cross-references to the tables in the SI.

In the analysis of material culture the authors have used presence / absence data to calculate Jaccard distances for cultural items. It was not entirely clear from the main part of the paper whether these are founded on the presence / absence of instruments or tools, or the presence / absence of distinctive traits found on the objects themselves.

Looking at the supplementary information, this becomes clear. However, without access to the actual contingency tables or computer code used in the analysis, it is not possible to access whether simple errors in the coding present in the S1 table (e.g. axe identified as 'ax', 'axe' and 'Axe'; there are other examples) influence the statistical results.

On ecological distance and biome type – the authors used the 'dataset' provided by Beyer et al 2020. This is not a dataset but rather a global vegetation model (which is based on BIOME4 equilibrium global vegetation model first used in experiments described in Kaplan et al. (2003)). The dissimilarities used to reconstruct the ecological 'distance' were therefore seemingly based on the number of different biomes contained within the territory of each group. I'd worry that this is merely reflecting the size of territory. It would be better to find some way of understanding what the real ecological 'distance' is between the various zones in the BIOME4 model.

Reviewer #4:

Remarks to the Author:

The study by Padilla-Iglesias analysed ethnographic, linguistic and genetic data of Central African rainforest hunter-gatherer (RHG) populations. Genetic data were collected from previous genome-wide studies of RHGs and then analysed together with cultural data (music instruments, subsistence tools and linguistic data) associated with them to infer their demographic history and to correlate cultural

and genetic exchange between these African groups. The study seems to be a comprehensive study of African RHG, however the manuscript needs clarifications about the genetic methods and the analysed datasets. I included detailed comments below to improve the clarity and text in the manuscript.

Major comments:

Figure 1 is lacking clarity. The authors analysed different datasets (music instruments, subsistence tools, linguistic, and genetic data), several results are presented for each dataset with different names for the populations/groups, and some populations are missing or have different names in the genetic dataset. It could be easier to understand the analysed populations/groups with two or more maps with the location of the populations/groups included in each analysed dataset.

It's also confusing that the name of the populations included in the genetic dataset does not match the names of the populations in the previous studies, making it more difficult to compare the results in Fig S1-S5 with results presented in previous studies (Patin et al. 2014; 2017; Perry et al. 2014).

As the authors and previous studies pointed out, there are genetic differences between Western and Eastern RHG. For the local ancestry de-convolution, the authors could explain the motivation to focus only on one "CAHG ancestry component" when they could also investigate the Western and Eastern ancestry components. In addition, in the section methods (page 19), the authors could indicate the sample size and samples included in the reference panel for each ancestry.

For comparisons between datasets and Mantel test results (Table 1), the authors could include heatmap plots for genetic distances (e.g., F_{st} for populations in each dataset), another heatmap plot for geographical distances (e.g., Km), and another heatmap plot for linguistic distances (e.g., PMI-weighted distance?).

The Discussion section should discuss more the genetic results presented in this study and put the new results in the context of previous studies. So far this section seems to discuss only the cultural results.

The manuscript could also comment on the exchange of musical instruments between Bantu-speaking populations and RHG, to explain the high diversity of instruments in Figure 4A in comparison with the subsistence tools in Figure 4B.

Minor comments:

Table S15, were all the studied populations included in the ASJP dataset? This supplementary table doesn't include all the RHG and a few names of Bantu-speaking populations. Could the authors better indicate the ASJP code of each studied population in the linguistic dataset? A heatmap plot of linguistic data could also help to understand the linguistic diversity of the studied populations.

For ADMIXTURE analyses. How many iterations were run at each K-group and whether a random seed was used each time?

The authors performed ADMIXTURE analyses from K=2 to K=8, however there are only suppl figures from K=2 to K=5. Other K0-groups could help to understand the population structure in Central Africa.

For local ancestry inference and masking, did the authors use any threshold to avoid miss-classify haplotypes?

For the phasing procedure, the authors could indicate whether they used reference panels and recombination maps. If that's the case, information about them is relevant.

In Fig S4 legend, the text indicated those are MDS results however in the plot the x-axis and y-axis are indicating: PCA1 and PCA2.

Discussion: change culture data for anthropological or ethnographic data, because linguistics is part of the culture.

Author Rebuttal to Initial comments

REVIEWER COMMENTS:

Reviewer #1:

Remarks to the Author:

The article 'Deep history of cultural and linguistic evolution among Central African hunter-gatherers' evaluates genetic, linguistic and material culture characteristics of 10-11 CAHG populations to understand the relationship between technology transfer between populations and the retention or adoption of different words in relation to geography, kinship and ecology. New methods for understanding relationships between genomic patterns and material culture (instruments) are proposed. The authors conclude that material culture and language patterns 'reflect long evolutionary histories'.

Back in 1977, Ian Hodder's 'The Distribution of Material Culture Items in the Baringo District, Western Kenya' (Man 12: 239-269) article about exchange of material culture in western Baringo really needs to be included at least tacitly in this discussion since it essentially arrived at the same conclusion albeit in quite a different way. The relationship between certain aspects of material culture, geography and language is a long-running discussion and it is well known that geography is a better predictor for material culture occurrences more so than language or kinship, but there are certain culturally specific attributes that are retained internally and others that are disseminated. Overall, there is nothing surprising about the results the authors present here, although some of the nuances of the relationship between language, geography and material culture are interesting (I do not buy into the genetic arguments made here at all; see below). I think that, if reframed a bit to incorporate modern anthropological thought about relationships between migration and material culture patterning, this paper could be publishable, although the changes would need to be rather significant.

We thank the reviewer for the comments. First, there seems to be a fundamental misunderstanding about the aim of our study. We are attempting to demonstrate the antiquity of cultural and linguistic exchanges in our species based on extant hunter-gatherers from Central Africa who have lived in that region for at least 120,000 years (Padilla-Iglesias et al. 2022). Hunter-gatherers are defined by a very distinct pattern of high mobility and fluid residence, which is ancestral to humans. Hodder researched East African pastoralists who emerged only in the Neolithic, and hence his conclusions cannot be fully informative of the demographic processes in a deeper time frame.

The reviewer seems to reduce our study to an attempt to show that geography is a good predictor of cultural similarities. Despite its many merits, Hodder's study of the relatively recent pastoralists in East Africa cannot tell us anything about the antiquity of material exchanges in Central Africa (or elsewhere), about the relationship between such exchanges and evidence of demographic contact among CAHGs and between CAHGs and more recent farmers, and about differences in transmission patterns of distinct domains (subsistence tools, musical instruments) as exemplified by the possibly most ancient cultural record in extant humans. In summary, as clearly stated in the manuscript, only a study of extant hunter-gatherers occupying Central Africa for at least the last 120,000 years can provide such insights into the early evolution of material culture and lexical diversification in our species.

However, we have now expanded our discussion of the existing literature in the introduction, comparing findings across other hunter-gatherer populations, as well as related methodologies investigating cultural evolution or genetic structuring elsewhere, following the reviewer's suggestion to further contextualise our study.

For example, we have added references to Migliano, 2017; 2020; Salali et al. 2016; Lewis et al. 2014 or Dyble et al. 2015 to stress the importance of researching the drivers of hunter-gatherer social structures and their consequences for different aspects of human behaviour in order to understand the implications the foraging niche occupied by our ancestors for unique aspects of human evolution.

In addition, we have added Hodder (1977) as a reference when discussing the rationale behind analysing different material cultural domains, alongside Derex and Mesoudi (2020) for a more recent account of demographic factors affecting cultural evolution. Similarly, we have also referred to Aguirre-Fernández et al. (2020) and Le Bomin et al. (2016) for more recent empirical quantitative approaches to the distribution of other aspects of music (the latter study also focusing on Central Africa).

In the discussion we have added references to contextualise our findings in relation to other studies focusing on hunter-gatherers. For example, we have included Polly Wiessner's (1983) survey on projectile point variation among San hunter-gatherer groups across the Kalahari Desert. Last, we have also extended our discussion of ethnographic remarks from CAHG themselves to further clarify the background to our research as well as the concordance of our findings with ethnographic remarks and historical accounts. For example, to support an ancient sharing of musical instruments and musical instrument words among the Aka and Baka, we have pointed the reader to the fact that rituals where these populations use similar musical instruments are also named the same in both populations, for example, to invoke the success of a quest (zòbòkò, ndàmbò, è.sà et mbèlà), to prepare for the death of an animal (mò.nzòli et kóbá); to celebrate the first capture of an important animal by a young man (mò.póndí)(Furniss, 2012). See also newly added references: Bailey et al., 1989; Hart & Hart, 1986; Blench, 2009 and Bahuchet, 1987; Bahuchet, 1995; Güldemann and Winkhart, 2020; Lewis, 2015; Cavalli-Sforza and Hewlett, 1982 for discussions on other topics.

First, the organisation of this manuscript is a bit odd. The Introduction gives away the whole pony before we really have a chance to evaluate the results or have a discussion. While it is common to have a sentence or two summary of the 'take-away' punchline at the end of an Introduction (i.e., thesis statement), I felt like I had already read everything I needed to know by Line 89. The rest is just supporting information. A lot of what is in the Introduction should be removed so that the data can be presented first and then the conclusions are made.

We have removed the summary at the end of the introduction and reworded it to stress the novelty in our study.

The figure captions are a bit wanton of information (Fig 1 also includes a grammatical mistake). Figures should be essentially self-explanatory with the captions, but the caption leaves too much to the imagination to be independently informative. See my specific critique of Fig. 1 below. Fig. 3 is good (captionwise, although the underlying assumption in the method is perhaps not). The other figures are not great or have missing bits of information critical to impart meaning.

We have modified figure captions. Please see main manuscript.

I am not a geneticist, but I have a decent background in statistics and I am not convinced that subtracting out genomic segments to correlate ancestry vs. different types of musical instruments is really a rigorous test of the hypothesis (L160-167). L179-180 reads more like a by-line from 'Guns, Germs and Steel' than

any modern anthropological perspective on material culture exchange. But, beyond that, I am not sure how one can exclude certain aspects of a group's genome and then correlate it specifically with instruments, either in the present or past. I read this section carefully several times and it makes no sense to me. Either the method is not explained well or I am missing some critical piece of knowledge to have this make sense to me, but, minimally I think some of the underlying assumptions need to be clarified. Are you looking at older instruments (i.e., from the archaeological record—from your Methods section it seems that you are not, but if not, then how can you connect 'old genomes' to specific instruments or names of instruments)? Assuming you can do black box subtractive math of genomes to identify sequential 'ur-populations' in the genome, how do you attribute specific instrument occurrences with those different segments given the known demographic complexity of Central Africa? To me, this is a step too far. In my opinion, this section of the paper should be dropped altogether because it makes no sense—at least to me.

We are surprised by the comment from Reviewer 1, since the feedback from the other three reviewers on our genetic analyses was very positive and supportive. We believe this is a matter of misunderstanding, especially given the suggestion of 'ur-populations' and 'old genomes', which are definitely not the outcome of neither genomic masking and IBD analysis.

As a summary: Genomic masking, as explained in the Methods section of the manuscript is a commonplace procedure in genetic analyses whereby the genomes of an "admixed" individual 1 (that is, an individual with ancestors from different populations) are compared with those of other "reference" individuals (representative individuals from each of the populations) in order to determine which genomic segments from individual 1 derive from individuals from each population. After the genome of individual 1 is decomposed into fragments belonging to each of the source populations, we can then remove all the genomic segments belonging to one of the source populations to perform our analyses only on the remaining segments of the population of interest.

In the present research, since we wanted to determine associations between material cultural evolution and deep CAHG population history independent of Bantu admixture, we used this technique to remove Bantu-associated DNA fragments from our sample, and in so, calculate genomic distances between populations based exclusively on CAHG genomic ancestry. Please see more details in the Materials and Methods section of our manuscript. For some examples regarding papers using a similar technique to study population-specific demographic history please see Schlebusch et al. 2020 or Ioannidis et al. 2020 and 2021 (pasted below):

Ioannidis, A. G., Blanco-Portillo, J., Sandoval, K., Hagelberg, E., Miquel-Poblete, J. F., Moreno-Mayar, J. V., ... & Moreno-Estrada, A. (2020). Native American gene flow into Polynesia predating Easter Island settlement. *Nature*, 583(7817), 572-577.

Ioannidis, A. G., Blanco-Portillo, J., Sandoval, K., Hagelberg, E., Barberena-Jonas, C., Hill, A. V., ... & Moreno-Estrada, A. (2021). Paths and timings of the peopling of Polynesia inferred from genomic networks. *Nature*, 597(7877), 522-526.

Schlebusch, C. M., Sjödin, P., Breton, G., Günther, T., Naidoo, T., Hollfelder, N., ... & Jakobsson, M. (2020). Khoe-San genomes reveal unique variation and confirm the deepest population divergence in *Homo sapiens*. *Molecular biology and evolution*, 37(10), 2944-2954.

The second step into our decomposition of genomic data into datasets corresponding to different time frames is the innovative approach we introduce in the present paper and involved the analyses and removal of IBD fragments from the remaining CAHG ancestry segments. IBD fragments are unexpectedly long similar DNA sequences due to a recent ancestor (in the case of siblings, their parents). When an IBD fragment is identified between individuals from two distinct populations, it allows the identification of a genetic sequence introduced into a population due to admixture. In that case, an IBD fragment represents a sequence (specifically, a consecutive sequence of SNPs) found in an individual in population 1 that is too dissimilar from the others, and too similar to sequences from another individual in population 2, to have evolved locally. In other words, it is a case of genetic similarity in sequence that cannot be explained by inheritance from the common evolutionary ancestor of the two populations. Hence, the sequence in pop 1 must have been introduced through admixture by a migrant from population 2. Despite the misleading name, an IBD is a fragment identical by descent from a recent migrant, not from the common ancestor of pops 1 and 2 (or ur-population).

The other feature of IBDs is that they don't last indefinitely. At first, an IBD fragment in one population will appear as a very long sequence identical to one in another population (the source of admixture); but every generation, due to crossing-over and recombination, the introduced chromosomes incorporate

fragments from the local background, and the size of common sequences (IBD size) between pops 1 and 2 decreases. After a number of generations, it is no longer possible to tell that the sequence was derived from a neighbouring group. For this reason, IBD analysis can only inform us of recent gene flow. In the case of humans, this is estimated to be about 200 generations. However, this also means that the size of the IBD between populations is inversely proportional to how long ago the migration/admixture event happened: a long shared sequence means very recent migration. Smaller sequences means longer time. Another consequence is that more recent migration may also mean more frequent migrations (as the most recent one may have been preceded by others).

To sum up: IBD fragments between populations tell us only about recent migration, not about early divergence of populations. Nothing about ur-populations. It tells us nothing about the split of Eastern vs Western CAHGs, or between any two populations in our study. This information is derived from classic phylogenetic methods. In addition, IBD fragments can only date recent migration. Even if there was migration prior to 200 generations, we would not know, as the evidence has been lost due to recombination.

How did we apply IBDs to our study? We applied a simple logic. If we identify IBD segments between all pairs of individuals belonging to different populations, we can single out parts of their genomes that derived from recent migration. Hence, if we calculate genetic distances between them with and without those IBD segments, we can remove genetic similarity that is due to recent contact between individuals from those two populations. Next, if we calculate shared musical instruments between two populations, we obtain a measure of cultural similarity in this domain. Calculating correlations between cultural and their genomic distances with and without IBD segments between distinct CAHGs, we can therefore compare whether there is a correlation between how culturally similar populations are and how genetically similar populations are, including or excluding genetic similarity that might come from migration. In our analyses, after removing genomic segments deriving from Bantu admixture from our CAHG genomes, we then removed IBD segments from the genomes of CAHGs. Hence, we knew that any shared IBDs between individuals from different CAHGs must have resulted from admixture between CAHGs.

We found a very clear correlation between their instruments and their genetic distance, with and without IBDs linking CAHG populations, indicating strong evidence that the distribution of musical instruments correlates with both ancient and recent patterns of migration between CAHG pops, before or after the Bantu. We found no correlations when the genetic distance still included Bantu sequences. In other words, the 'dating' comes naturally from the fact that we know when the Bantu arrived, and we can identify the genetic evidence of this arrival, and exclude it. As the reviewer can see in the main text, we repeated the procedure to subsistence tools etc. But we believe that there is no black box in any of what we proposed.

We have added the clarifications above to the SM.

In sum, the paper uses a lot of statistical methods, many of which are rather complex, that compound upon each other to arrive at a rather simple conclusion. Particularly by the time the genetics arguments are laid out, I felt a bit lost in the forest of methods that seemed to teeter on assumptions that could not be fully justified. I am familiar with all of the methods, at least in principle, that were used in the paper and actually teach statistics modules to archaeology Masters' students at a university, so it is not that I am averse to complex math or the use of statistics. It is just that there needs to be a balance of statistics along with good old-fashioned ethnography, standing on decades of research of anthropologists going into the field and interviewing people about what, when, where and why people make economic decisions related to material culture. In my mind, it seems that this paper has arrived at the most parsimonious conclusion to the question they set out to answer, but the way they get there is not clear and is a bit of a house of statistical cards that needs more solid footing.

Although the reviewer says we reach a 'rather simple conclusion', we never saw this conclusion clearly stated in the reviewer's comments. As explained above, the reviewer seems to suggest that our article primarily attempts to reinstate a known link between geography and culture. But our title ("Deep history of cultural and linguistic evolution among Central African hunter-gatherers") clearly suggests otherwise.

We present evidence that musical instruments have been shared among CAHGs across central Africa and over long distances before and after the Bantu arrival, that subsistence tools follow a very distinct pattern of distribution and relationship to ecology and migration, and that music-related lexicon precedes the borrowing of Bantu languages. For sure, none of those evolutionary conclusions are found in Hodder's study of pastoralists for example. It should be noticed that most cultural data in our article were derived from original ethnographies. We stand by our procedures to analyse such a rich database, and believe that our results are a valuable addition to the literature on the origins and diversification of human culture.

Minor matters:

Fig 1: Does not have a scale. Nor is there a legend to indicate what the different shades of green represent. The use of modern national borders should be clarified (although it isn't particularly pertinent to anything in the paper), and if included, you should have country names labelled.

We believe it does make sense to include contemporary national borders as these are contemporary populations, and therefore, it helps readers not specialised in Central Africa to contextualise our research. The caption in the figure explains what the shades of green represent. We have added country labels and a scale in order to follow the reviewer's advice.

Fig. 4: I know that the ethnic group names were already presented in Fig. 1, but in the discussion below Fig. 4, I had to go back and forth between Fig. 1 and 4 to associate geography with the groups being mentioned. This is not an effective presentation style.

We added Fig 1 specifically to provide the additional information on populations etc. in our opinion, having the names in Fig. 4 would distract attention from the main results. No other reviewer raised this concern.

L105: Why 'acculturation', specifically? You imply a dominance hierarchy in which farmers enculturate foragers, but this is not always the way it works. There is often a two-way exchange of material culture (see Hodder's 1977 article) that is not based on 'acculturation'. Please choose another word here or expend words to explain why it is 'acculturation' as opposed to other mechanisms of social exchange.

We have replaced this term with "cultural borrowing" from farmers.

L148, 208, 237, 471: No comma needed.

Removed. Thanks.

L253: Oral traditions ≠ legends.

We agree. However, Arom and Thomas (1974) or Bahuchet and FURNISS (1995) discuss "legends" from different populations that attribute the origins to several Central African instruments to hunter-gatherers. We chose to use their term.

L256: ‘an’ should be ‘a’; and, see again, Hodder (1977) among many sources that demonstrate that linguistic connections do not map clearly over genetic affiliations or material culture. We don’t know when these words originated or what the context is for their shared phonemical linkages. We only know that they exist across space irrespective of language similarities. The implied connection between language (often used as a proxy for kinship since languages are socially learned) and material culture hearkens back to some old thoughts about migration and material culture popular in the 1970s through 1990s, which I advise the authors to contextualise better than they have done here.

- See also <https://doi.org/10.1080/0067270X.2011.629525>
- See also <https://doi.org/10.1111/j.1467-9655.2010.01674.x>
- See also <https://doi.org/10.1038/s41598-018-31123-z>

It is precisely through the comparison of shared linguistic terminology, shared culture and shared genomic ancestry at different time depths that we estimate when the words originated. This is more so the case where the shared words between CAHG groups are not present in the Bantu populations from which the CAHG inherited their languages.

Regarding the context of their shared phonemic linkages, we do know the context for those links, and have tried making them more transparent by including Prof Balthasar Bickel and Bogdan Pricop in the manuscript, who have assessed the shared phonemic linkages between each pair of terms within the context of the grammars of their respective languages, and thus determined whether such phonemic linkages were most likely the result of orthographic conventions, noun class marking, etc. We could determine whether putative shared specialised vocabulary is indeed shared due to common descent (independent of the recent borrowing of CAHG of farmer languages) or not. Please see the updated Materials and Methods section as well as the supplementary text for additional explanations on how this was done, as well as the newly included Dataset S2 with a detailed explanation of the criteria used to determine whether each of the terms was shared or not, and the context for the shared phonemic linkages.

We thank the reviewer for the comments and hope that our reply has clarified the purpose of our study and our genetic and linguistic methodologies.

Reviewer #2:

Remarks to the Author:

This article compares the evolution of different cultural traits, namely musical instruments and foraging tools, as well as the lexical terms for both, along the deep and shallow history of 10 Central African hunter-gatherer (CAHG) groups. This evolution is compared with the genetic and linguistic history of these groups, as well as its ecological and geographical situations. The results of this study show that both the musical instruments and their names present in these CAHG groups in current times are the product of their deep history, predating the Bantu expansion and language shift of these groups, as well as the interaction between different groups. On the other hand, foraging tools, as well as their names, would have been independently adapted to the different ecological regions. This is an interdisciplinary study, including genetics, anthropology, linguistics and ecology, with a focus on genetics, where a novel method is implemented for studying different layers of history.

Although the results are interesting, both for specialized scholars and for more broader audience, I see flaws in the presentation of the methods and results, for which I would recommend major revisions before the manuscript is accepted. Below, I list 4 major comments, as well as several minor comments to be addressed. I must say that my background is in quantitative analyses and linguistics, for which my comments on the genetic analyses are not very specific, barely educated guesses.

Major comments

1. Code availability

The Data and code availability section states that “All the data and code required to reproduce all the results reported in the manuscript and supplementary information will be made available upon acceptance of the paper in the following Github repository: <https://github.com/ceciliapad/GenCultEvo”>, which is private at the time. Therefore, I wasn’t able to review the code, only the Dataset S1. Please make it available to allow for replicability.

We are sorry about this. Indeed, we should have made the repository available for the review process. Upon acceptance of the manuscript, it will be made available for the general public as well, but in the meantime, reviewers have special means to access the data and code in the following link: <https://www.dropbox.com/scl/fo/gj4qg3wp73jfovpm9hkn3/h?rlkey=uml9a3js1drmcwec6fkwxrwqu&dl=0>

We have indicated the instructions in the manuscript file as well. Also, all genetic data are in the public domain.

As for the Dataset S1, I detected some inconsistencies, as having up to four different terms for one single trait: (arrow, Arrow, arrows, Arrows). These issues might have been solved in the code by re-coding all these forms into the same value, but I cannot evaluate that without seeing the code.

Thanks for that comment. Dataset S1 was not automatically processed for analyses. Another manually-made table with presence/absence of each item each population was used instead in the analyses (please see the repository for a copy). Hence, despite slight variation (which referred to the nature of the image/original terminology used by ethnographers) in the labelling of items in Dataset S1, any population having for example “arrows, Arrow, Arrows etc..” was coded as having arrows present. However, we have nonetheless updated Dataset S1 for consistency according to the reviewer’s suggestion.

2. Language classification and linguistic distances

I have several comments on this section, regarding the different language classifications presented, the method used for defining the linguistic distances, and finally the cognate judgment used to decide whether two lexemes coincide across languages for a given concept.

2a. Different language classifications (Tables S12-13)

I don’t understand the rationale for including two different classifications of the languages involved (one of them including the speculative Niger-Kordofanian macrofamily), none of which will be ultimately used for calculating linguistic distances –distance between wordlists is used instead, see next point. Also, glottocodes, one of the 2 standard labels together with ISO 639-3, are not reported (neither the Glottolog classification mentioned), and the labels used for denoting languages in the first Table S15, where the linguistic distances are reported, are different from any of the classifications mentioned, making the match between them hard to follow.

I tried to link the languages in Tables S12, S13, and S15 (first) but wasn’t able to do all the matches. For example, I don’t understand if the languages for Mbendjele and Aka are supposed to be the same or not (if there is no typo, the Mbendjele speak the language with ISO-code mgd, meaning Moru (Central Sudanic), but I imagine it meant ngd instead, reducing the total number of languages to 13 instead of 14).

Thanks for spotting the issues. The Mbendjele and Aka speak two dialects, of which the closest language is Ngando (ngd, not mgd; thanks for spotting the typo). This is why they were classified in the table as “CL” (i.e. the language for which we have information is the closest farmer language classification-wise). However, since the Mbendjele were not included in the analyses of material culture, we have removed them altogether from the article.

Secondly, the ISO-code kin corresponds to Kinyarwanda. Was this meant to refer to Twa (Glottocode twaa1238, variety of Ljenje, Glottocode lenj1248, ISO-code leh)?

Regarding the *kin* code, we are sorry for the confusion regarding the language name, but the ISO-code is correct. Kinyarwanda is referred to by the ASJP database as “Rwanda Twa Kinigi” (kiny1244 in Glottocode). As indicated by Bahuchet (2006) and the Ethnologue classification, its closest farmer language in the classification is “Rundi”.

	Rwanda Twa Kinigi	kiny1244	kin	kin	-2.00	30.00	no	no		NC, Bantu	Atlantic Congo, Volta Congo, Benue Congo, Bantoid, Southernbantoid, Narrowbantou, Eastbantou, Northeastsavannabantou, Greatlakesbantou, Westernlakesbantou, Kivu, Westhighlandskivu
--	-------------------	----------	-----	-----	-------	-------	----	----	--	-----------	---

Finally, I don’t see a match for the languages NZEBI and SANGO in Table S15. The first seems to refer to the glottocode nzeb1234, which is a branch containing 7 languages according to Glottolog, and Sango has glottocode sang1328, ISO-code sag, not present in Tables S12-13. Please specify the correct matches, so this is clear for the reader (consider the table below just as an attempt: the authors certainly know better than me what exact languages they mean and will be able to produce a much better matching).

We have now added the Glottocodes of all languages for additional information and clarity. We have also relabelled Table S3 with the Glottocodes so that it is clear which language corresponds to which.

2.b Linguistic distances

I disagree with the method used here to calculate linguistic distances (Lines 227-246 and 552-560). Although proven to align with known phylogenetic classifications to a reasonable extent in Jäger 2018 (ref 46), this method to measure linguistic distances is non-standard, and has several caveats.

It is based only on the distance between wordlists, so it doesn't necessarily correlate with phylogenetic distances (which is what I assume is the authors' intention). In creating a phylogeny, not only pairwise-lexical distances are measured, but also different possible evolutionary trajectories are considered, and archaeological data can be used for calibrating the dates. If the matches in the table above are correct, EFE should be by far the language more distant to all the rest (non-Atlantic-Congo), followed by BAKA (Ubangian), TIKAR (Northern Bantoid), and then the other 10-11 languages, closer to each other than to any of the former three. However, from Table S15 we can see that e.g. NANINGI is closer to EFE than to BAKA, and almost as far to TIKAR and even GYELE (both Bantu languages) as to EFE. The same is true for NGANDO, TEKE, and TWA, which are closer to EFE than to SANGO. Also, KOYA is closer to BAKA than to SANGO, TIKAR, and BILA, etc.

We understand the concern by the Reviewer. The motivation for calculating linguistic distances using words was that we needed a baseline measure of lexical similarity between languages. Our question here was not the precise phylogeny of Central African hunter-gatherer languages, but instead: can the processes responsible for the distribution of lexical diversity across Central African Hunter-gatherers also explain the distribution of musical instrument terminology, or do we need additional/different processes to explain the latter?

In other words, we understand that the level of linguistic differentiation between language pairs may not correspond to their phylogenetic distances or time from split. But for our analyses, what matters is whether musical instrument or subsistence tool lexicon is differentiating at the same level or not.

Nonetheless, PMI has been shown to be informative for performing distance-based phylogenetic inference (Jager, 2018), besides providing a measure of general vocabulary similarity between CAHG groups. As further evidence, to determine whether PMI distances were good indicators of phylogenetic distances within our sample, we performed a correlation test between PMI distances between groups and patristic distances (as you suggest in the next comment), and the correlation was 60% ($S=5353.1$, $p<0.001$). Therefore, it seems that both generally and within our language sample, PMI distances are good indicators of phylogenetic distances.

We also wish to note that the use of PMI distances from the ASJP as proxies for phylogenetic distances specifically for analyses relating cultural evolution to genetic evolution is an established approach. See for example Matsumae et al. 2021 *Science Advances*. Nonetheless, see the response below to the following comment by the reviewer for the inclusion of similar calculations using actual phylogenetic distances.

Reference:

Matsumae, H., Ranacher, P., Savage, P. E., Blasi, D. E., Currie, T. E., Koganebuchi, K., ... & Bickel, B. (2021). Exploring correlations in genetic and cultural variation across language families in northeast Asia. *Science Advances*, 7(34), eabd9223.

Since all (but one) of these languages belong to the Atlantic-Congo macrofamily, the ideal way to measure linguistic distances here would be by using the patristic distance (time elapsed since the most recent ancestor for the 2 languages) from a known phylogeny. The phylogeny by Koile et al. 2022 (R1) (please add the reference, see below), in its augmented tree, includes all 10 Bantu languages of the 13 languages in Table S13 (see table above), only missing Baka (Atlantic-Congo > Ubangi), Tikar (Atlantic-Congo > Northern Bantoid), and Efe (Central Sudanic). The phylogeny by Grollemund et al. 2015 (ref 66), includes 8 of these. The distances for the 3 missing languages can be approximated with a higher distance for the two Atlantic-Congo non-Bantu languages to all Bantu, and a further higher one for the Central Sudanic language.

A simpler option (although less precise) would be to part from an accepted classification of groupings such as Glottolog (R2), and take as distance the amount of nodes between a pair of languages (plus a fixed quantity for the case of Efe vs all Atlantic-Congo languages).

(R1) Koile, E., Greenhill, S. J., Blasi, D. E., Bouckaert, R., & Gray, R. D. (2022). Phylogeographic analysis of the Bantu language expansion supports a rainforest route. *Proceedings of the National Academy of Sciences*, 119(32), e2112853119.

(R2) Hammarström, Harald & Forkel, Robert & Haspelmath, Martin & Bank, Sebastian. 2022.

Glottolog 4.7. Leipzig: Max Planck Institute for Evolutionary Anthropology.

<https://doi.org/10.5281/zenodo.7398962> (Available online at <http://glottolog.org>, Accessed on 2023-05-04.)

Thank you very much for this suggestion. In order to validate and supplement the findings using PMI distances, as advised by the reviewer we calculated patristic distances between all Bantu languages in our sample (see Fig. S16) using the augmented tree from Koile et al. 2022. For groups speaking non-Bantu languages, we set the distance between them and all other languages to be the maximum possible distance in the tree.

In addition, we also calculated linguistic phylogenetic distances using the Atlantic Congo and Central Sudanic Glottolog trees (as suggested by the reviewer) and counting the number of internal nodes between each pair of CAHG languages (used as tips)(See Figure above, which has also been included as Fig. S15). The correlation between the three linguistic measures was between 60-80%, indicating that all three methods carried a heavy signal of phylogenetic linguistic relatedness.

We then used the three measures of linguistic distances to run Zero-Inflated Poisson Models predicting the sharing and number of shared specialised vocabulary items between CAHG groups and our conclusions remain unchanged, as we now mention in the main text (See Tables S14-S19). Groups speaking phylogenetically closer languages were not more likely to share musical instrument words, whilst groups that were genetically closer (i.e. more related) were, as well as those living spatially closer. These findings are identical to those exploring the predictors of sharing musical instruments themselves, and held true both when considering all shared musical instrument terms as well as those unique to CAHG. This suggests that the evolutionary dynamics shaping the distribution of musical instruments across Central African hunter-gatherer groups are the same as those shaping the distributions of the terms used to designate them.

2c. Cognates judgment and comparison with farmers' languages

How was the cognate judgment carried out? (Lines 200-218). Although it is mentioned later in the text that only exact matches were considered (Line 539: "...we only counted as "shared" words that were identical [after removing prefixes and suffixes]."), in Table S14 there are cases of different scripts (e.g. ṅòṃ / ngom) and presumably small variation (e.g. ruma / baruma, sànze / sanzi, likembe / kembì), while other items are considered as different cognate sets although they might be similar enough (e.g. how is ngomo / ngoma / ṅòṃ / ngomu / ngom different from ṅòṃbì / ngombì? I'm referring exclusively to the form here, not to the semantics, which seems to denote a different instrument). Was this checked by an expert in these languages?

Secondly, in Line 532: "Note that the same instrument may be designated by more than one term within the same cultural group due to within-group linguistic heterogeneity and the fact that the same instrument might be named differently depending on the occasion." What has been the criterion to select one form over another in these cases? Please specify.

Finally, in Lines 312-318: "We have also shown that although CAHGs have adopted languages from farmers, music related words were most likely inherited from ancestral populations together with the

corresponding musical instruments. By contrast, the fact that very few tool-related words are shared between CAHGs may reflect either a convergent origin of the similar tools in groups with divergent languages (and hence distinct words for similar tools); or an ancient common ancestor to both subsistence tools and tool-related words, with tool (but not word) differentiation over time being limited by ecological constraints.” In order to be sure about this, it would be important to compare these words with the ones for the same musical instruments (if present) by the farmer (Atlantic-Congo of Central Sudanic) speakers of these same or related languages.

Thank you very much for this remark. Indeed, previously we had been extremely conservative and included only words that were identical in two hunter-gatherer languages after removing well-known prefixes and suffixes. But following the reviewer’s advice, we included in the project two linguists expert in African languages (Prof. Balthasar Bickel and Bogdan Pricop) who reassessed our criteria to establish cognate sets. They relied on the use of grammars of each of the languages of CAHGs and their neighbours (see updated methods section, and supplementary Dataset S2). We believe this greatly improved the reliability of our analyses, and allowed us to confidently expand our list to include all the words deemed as cognates (variation only due to orthographic changes, changes in noun class system), after consulting the grammars of the respective languages. We remained conservative and therefore, cases where the origins of words not sufficiently backed up with evidence were identified and removed from the most stringent analyses (including Figure 4 in the main text).

In addition, also after consulting (where possible) the grammars of the CAHG languages and of their neighbouring farming populations, we noted whether (1) the objects in our list were unique to CAHGs, (2) whether the words denoting the objects were unique to CAHG (as opposed to shared with their non-hunter-gatherer neighbours), and (3) whether there was unequivocal evidence that the words denoting the objects were shared between CAHG groups. Then, we divided the words into two categories: words shared for a particular object between 2 or more CAHG groups, and shared words exclusive to CAHG and where the evidence of the sharing of terminology was unequivocal. We then run all analyses with shared words from both datasets. We believe both datasets are important as CAHG are known in Africa for their extensive music and musical instrument repertoires. Hence, many Bantu (and other farmer) populations rely on them to sing and dance during their own ceremonies and rituals. This often results in ethnographers reporting that farmers use the hunter-gatherer word for particular instruments, as the instrument comes from the hunter-gatherers and is often only used in contexts where the hunter-gatherers are involved (see Fürniss, 2012; Fürniss and Bahuchet, 1995 for more details). See Table S13 for the final list of shared words.

In addition, we have included now Supplementary Dataset S2, a full spreadsheet with all the shared words, notes on the presence of the objects and words in neighbouring farming populations, and in the

African continent, as well as the evaluations by linguists to assess cognancy, plus the sources of the grammars used to support those judgements.

We have also included a section in the main manuscript explaining this as well as Supplementary Text 2, with further details and references. We have also modified Fig 4 to include only those shared words that were unique to CAHG populations.

3. Correlation analyses

Tables 1, S5, S6, S7, and S9 test similar hypotheses, namely correlation between the variables of interest: genetic diversity, geography, ecology, and cultural traits (musical instruments and subsistence tools), by using different statistical methods, namely Mantel test, CADM or Multiple matrix regression. I think it would be more clear if when the same comparison is performed with a different method, both results are summarized together in the same table or plot. Otherwise, it is not clear why three different methods are used to answer the same question, and in which cases they agree or disagree. This is explained to some extent in Lines 489-524, but it is still somehow confusing why so many analyses are attempted on the same data.

Thank you for this remark. Whilst Mantel tests test for the association between two matrices (and therefore provide a statistical assessment of whether the two are significantly correlated), multiple matrix regressions can control for a third (potentially correlated, such as in our case geographic and genetic distances - see Tables S5 and S8). Last, CADM assesses support for the hypothesis that a particular distance matrix (or combination of distance matrices) is significantly different from another matrix (or combination of matrices). Therefore, results cannot be included in the same table, as the models do not include the same predictor and response variables. Nonetheless, in conjunction, they can be used to assess the relative strength of the correlations between geographic, genetic and cultural distances.

Most importantly, all the results were congruent, as geographic and genetic distances (when based exclusively on CAHG ancestry components) were shown to be strong predictors of cultural distance based on musical instruments (in Mantel tests and CADM tests), both variables being equally strong predictors of cultural distance, and significant when controlling for ecological distance but not when controlling for one another in multiple matrix regressions. On the other hand, for subsistence tools, ecological distance was a significant predictor of cultural distance (in Mantel tests), even when controlling for geographical distance (but not for genetic distance).

4. Pseudo-coordinates

I am not familiar with the transformation into pseudo-coordinates (Lines 137-142 and again Lines 258-260). This said, phrases like these are confusing to me (Line 258-260): “Moreover, a SpaceMix analysis showed no clear visual match between geographic distances and pseudo-distances estimated from subsistence tools (Fig. S6).” Is this analysis solely based on visual match of geographic distances with pseudo-distances retrieved from the dataset of interest? If this is the case, I don’t see how Fig S6A for musical instruments shows a clear match, while Fig S6B for subsistence tools shows a clear mismatch. They don’t look that different to me. In Materials and methods, the only explanation of the method is “The geno-geographic and cultural-geographic coordinates generated were compared with the actual geographic coordinates.” Could you please specify how this comparison is quantified?

We thank the reviewer again. This was our mistake. Fig. S6 was mislabelled in the former version of the manuscript. In fact the left panel corresponded to subsistence tools, and the right panel to musical instruments. We have now remade the legend (see new Fig. S12)

Also, we have now provided a quantification of the Euclidean distances between the geographic coordinates and pseudo-coordinates (based on musical instruments, subsistence tools and genes) of the 10 CAHGs (See Table S10). We can now estimate correlations and regressions between them and properly quantify their possible ‘match’. For example, the distance between the geographical locations and pseudo-coordinates estimated from musical instruments was generally smaller (for 7/9 CAHG groups) than those between the geographical locations and pseudo-coordinates estimated from subsistence tools ($W=20$; $P=0.038$)(see Fig S18). This means that the structure of variation in musical instruments mapped better onto the geographical distribution of groups. We have added the results and statement above to the main text.

Minor comments

General

Please check the labels of the supplementary materials: Table S4 is non-existent, and the following tables' numeration is shifted (e.g. Tables S6 and S9 referred to as Table S5 and S8 in line 111, among many other cases). Also Table S15 is the name for two different tables, please correct. Also, please refer to the supplementary materials in order. It is hard to navigate them in the current order (e.g. the first supplementary figure referred is S1, then S10, S11,...).

Thank you very much for your careful read. We have relabelled all tables and figures in the supplementary material so that their numbering matches the order in which they are referenced in the main text.

Introduction

Line 41: "CAHGs" is used for Central African Hunter Gatherers, but it is not defined before in the main text (only in the abstract).

We have now included a definition of CAHG the first time the term is mentioned in the introduction.

Lines 43-45: The abstract mentions 10 CAHG groups, but this sentence mentioned 11 groups, speaking 14 languages. I understand that there is at least some data (cultural) for 11 groups, of which genetic data is present only for 10, but this is not completely clear in this first paragraph, and it makes it look inconsistent with the abstract, please clarify. Also, the 14 languages are not addressed either in the figure or in the table mentioned, and it is unclear e.g. to understand which groups are multilingual. Tables S12-13 do mention languages: There are 14 rows (in Table S12), but the languages for Mbendjele and Aka seem to be the same, reducing the list to 13 languages, is this correct? Table S13 has 13 rows, but again Mbendjele and Aka seem to have the same language, and one of the languages of Batwa (East) is missing.

We agree that this was confusing. Since the Mbendjele were not included in the analyses of culture and speak a dialect of the same language as the Aka, we have now removed them from the linguistics analyses to avoid confusion. Now in general we refer to 10 groups.

Line 59: please add one more relevant reference, on the use of musical instruments for studying the human history: (R3)

(R3) Aguirre-Fernández, G., Blasi, D., & Sánchez-Villagra, M. (2020). Panpipes as units of cultural analysis and dispersal. *Evolutionary Human Sciences*, 2, E17. doi:10.1017/ehs.2020.15

Thank you for this. This reference was in an initial draft of the manuscript and indeed it is highly relevant for the discussion raised in the paper. It has now been added to the main text.

Line 61-63: Consistent spacing between magnitude and unit: “160 km” in line 61 but “2,000km” in line 63.

We have now corrected this.

Line 73: groups referred to as Biaka and Mbuti here are referred to as Aka and Mbuit respectively in the caption of Figure 1. Please rename consistently.

Thank you very much. We have corrected this

Figure 1:

Map: The colors chosen for single and multiple genetic groups are shades of green, while the background map uses green for the steppe. In addition, international borders such as those inside the Baka group are marked with the same color and style as group borders (e.g. between Aka and Mbendjele). Both these design decisions make the map hard to read, and can be solved by e.g. replacing the background map for one with different colors and international border style (or no international border at all). These comments also apply to Figure 4 and S14.

We believe leaving international borders in Fig.1 helps contextualising the area inhabited by CAHG (we have now added country labels as suggested by another reviewer), and the geographic extent of each of the groups. Nonetheless, we have changed the colour of international borders within group territories to avoid confusion.

Caption: Central African hunter-gatherer → gatherers. Also, check the names used for Aka and Mbuit (see above in Line 73).

Thank you very much. We have corrected this,

Results

Line 103: Add relevant reference on the Bantu expansion and their interactions with CAHG: (R4)

(R4) Klieman, K. A. (2003). " The pygmies were our compass": Bantu and Batwa in the history of west central Africa, early times to c. 1900 CE. Greenwood.

Indeed this reference is relevant, and we have thus included it in the paragraph.

Line 103 (and also Lines 275–283): “We calculated Jaccard distances between populations...”. Please refer here to Materials and Methods → Measuring cultural distances. Also, if I understood it correctly, only attested cases were considered. Therefore, were NA’s considered the same as absences? This distinction is crucial when drawing conclusions on a highly sparse set such as subsistence tools (Lines 275–283). Please clarify in Materials and Methods.

Hopefully the fact that the data made available in our public repository will increase the transparency of our analyses and datasets. As you can see in their respective datasets, there are only 2 NAs in the musical instrument dataset, and none in the subsistence tool one. Nonetheless, a main feature of Jaccard distances is that they only count shared presences as “sharing”, and are thus widely used in archaeological studies where comparisons are between evidence of presence vs. absence of evidence (rather than evidence for absence). See for example Timbrell et al. 2022; Blinkhorn and Grove, 2021; Shennan et al. 2015. This is the reason why we are running correlation tests using Jaccard distances.

Line 106: NeighborNet algorithm, please add reference: (R5)

(R5) David Bryant, Vincent Moulton, Neighbor-Net: An Agglomerative Method for the Construction of Phylogenetic Networks, *Molecular Biology and Evolution*, Volume 21, Issue 2, February 2004, Pages 255–265, <https://doi.org/10.1093/molbev/msh018>

We have now added this reference.

Figure 2, caption: "...PCoAs for based on..." → delete the "for"

This has been amended accordingly.

Line 121: Is (Fig S3) correctly referenced?

We have relabelled all tables and figures in the supplementary material so that their numbering matches the order in which they are referenced in the main text.

Line 122 (also Line 135 and 254): see Methods → see Materials and Methods

This has now been corrected accordingly.

Line 126: "We found a correlation..." → "We found a significant correlation..."

This has now been corrected accordingly.

Line 128: Table S5 includes the Mantel test, while Table S6 includes the CADM analysis (please correct labels and add the missing one).

All the labels and references to SM Figures and Tables have been checked and ordered so their numbering matches the order in which they are referenced in the main text.

Figure 3: I don't understand the information conveyed by this figure (I must say here that I don't have a background in genetics, though). Is this merely an illustration of how the genetic sets look before and after removing the segments associated with each ancestry layer, or should it show any further

information? Having 10 and 12 columns in each step, respectively, I imagine each of them is associated with one CAHG group? What do the heights stand for?

The figure is a schematic representation of the human karyotype (i.e. the human chromosomes - which have lengths corresponding to the ones shown in the diagram, although this is not relevant as it is just a schematic representation). We have now updated the legend to clarify this. It is just a diagram to illustrate the two way masking process by which we identified and eliminated genomic segments across the genome corresponding to different ancestry sources. In addition, we have modified the figure to include the pink segments in the initial panel to show that it includes all genomic segments.

Line 173: “The approach implies various predictions” → “The approach can result in various different outcomes”

We believe those are predictions, and not outcomes, as we are presenting hypothetical scenarios,

Line 202: “Hence, either CAHGs adapted words from farmer languages to name their instruments, or preserved the original names from their lost languages.” These are not the only possible options. There could have been innovations for these lexical items in (some of) the CAHG languages either before or after the adoption of the farmers’ neighbouring languages.

We agree and adapted the text to account for other options.

Figure 4: Instrument 1 is called ikembe here, but referred in the rest of the text as in Table S14 as kembe.

Thanks. Figure 4 has been modified and now includes only shared terms between CAHG groups that are unique to these populations, and where there is no uncertainty/ambiguity as to whether the words are shared or not.

Line 245: they → their

This has been modified accordingly.

Discussion

Line 319: “We believe that extending our procedure may lead to the identification of other ancient shared linguistic terms in cultural domains known to precede the arrival of farming groups.” What other cultural domains do you refer to?

We have now added examples: vocabulary related to forest, forest life and rituals regarding the forest as well as kinship terminology and rituals and a reference to Bahuchet, 1987, who follows a similar reasoning to ours (comparing sharing of specialised vocabulary between Aka and Baka across cultural domains vs. the sharing of general vocabulary)..

Materials and methods

Line 340: data provided by any one source → data provided by every source

This has been modified accordingly.

Line 345: it refers to N=1444 material cultural records, but the Supplementary Data S1 includes 1370 datapoints.

Sorry about this. Indeed the initial dataset had 1444 records from which 74 were excluded due to unreliability on their precise geographical location.

Line 356: and creating partial polygon objects → and partial polygon objects were created

We have modified the sentence accordingly but please note that we write “spatial” not “partial” polygons.

Line 358: “Note that due to data availability, for the analyses concerning cultural repertoires, we did not distinguish between the Aka of the Central African Republic and the Mbendjele of the Republic of Congo, as genetically and culturally these groups are often considered a single cultural group”. Does this also apply to Sua and Efe?

The reviewer is right that Sua and Efe are also many times considered together as Mbuti. However we did have access to separate cultural data for the two groups. Therefore we have not merged them for analyses.

In the case of the Mbendjele-Aka, genetically they represented a single population and culturally they are often considered a single group as well. Therefore, even if they speak different dialects, we consider them as a combined group in all our analyses and have omitted Mbendjele from the map.

Line 396: between iteration → between iterations.

This has been modified accordingly.

Line 408: “As recommended, we used 15% of the data for training set 1, 80% for training set 2 and 5% for validation...” As recommended by whom? Is this the standard practice in the field? Add reference if relevant.

Thanks. This was recommended by the developers of the software, which are also coauthors in the manuscript. We have added this remark and referenced the paper. We have also included as Table S22 the confusion matrix produced that reveals the high performance of Gnomix on the validation set with the chosen configuration.

Lines 421-423: “2,500B.P. is a key date for the purpose of this study given that it is thought to represent the initial north–south migration of Bantu speaking communities across the Equator following a fragmentation of rainforest that would have created corridors for human migration.” Please mention that recent studies have cast doubt on this, e.g. reference (R1).

(R1) Koile, E., Greenhill, S. J., Blasi, D. E., Bouckaert, R., & Gray, R. D. (2022). Phylogeographic analysis of the Bantu language expansion supports a rainforest route. *Proceedings of the National Academy of Sciences*, 119(32), e2112853119.

Indeed Koile et al’s paper argues for an earlier crossing of the rainforest before the creation of the Sangha River Interval corridor 2,500BP. We have now included this important reference, and wrote a sentence about it in the corresponding section.

Line 458: decomp-posed → decomposed

Done

Line 474: Biome composition was obtained from reference 71, but this reference reconstructs the biomes for the last 120,000 years. At what time(s) was it obtained?

This has been modified accordingly. We obtained the biomes for the present.

Line 476: Beyer (82) → Beyer (71)

We have now solved and checked this problem with the formatting of the references.

Lines 506 and 512: `CADM.global` and `CADM.post` are not names of statistical methods, but rather the function names for a specific implementation of them. Please clarify the reference, possibly (R6)?

(R6) Paradis, E., & Schliep, K. (2019). `ape 5.0`: an environment for modern phylogenetics and evolutionary analyses in R. *Bioinformatics*, 35(3), 526-528.

We now specify this in the section “assessing the drivers of cultural distances”, where we define “Congruence among distance matrices” and introduce the `CADM` acronym as well as “global test of significance” that we specify is implemented with the `CADM.global` function, and last “*a posteriori*

tests”, that we implement with the function `CADM.post`. We also added a reference to the paper by Legendre and Lapointe (2004) where the method is introduced. We implemented these functions using the *ape* package and we have thus now included the reference to it suggested by the reviewer.

Reference:

Legendre, P. & Lapointe, F.-J. ASSESSING CONGRUENCE AMONG DISTANCE MATRICES: SINGLE-MALT SCOTCH WHISKIES REVISITED. *Aust NZ J Stat* **46**, 615–629 (2004).

Line 536: `calleg` → `called`

This has been modified accordingly.

Lines 548-551: “To calculate an estimate of linguistic phylogenetic distance between pairs of CAHG languages, we obtained the ISO 639-3 code of either the CAHG language itself or the farmer language which the CAHG dialect was derived from.” The last expression, implying that dialects “derive” from (larger, more prestigious) languages might sound harsh to the linguistic audience. I would rephrase it to something like: “...we obtained the ISO 639-3 code of either the CAHG language itself or the farmer language most closely related to it.”

We thank the reviewer for this suggestion and have modified the manuscript accordingly.

Supplementary materials

Fig S1-S2: Please improve the resolution of the text (it is hard to distinguish e.g. Baka vs Biaka).

We have now done this and changed the Biaka name to Aka to be consistent between cultural and genetic analyses.

Fig S2: In caption: (left) and (right) should be (top) and (bottom) respectively

We have now modified this.

Fig S3-5: It might be hard to distinguish the different colors for some readers. It would help to put the labels corresponding to a population near the corresponding dots. Also, at least in Fig S3, it would help to visually mark which groups correspond to NW, SW, and E (e.g. with an ellipse with dashed contour including the relevant groups in each region).

Thanks for the suggestions. However, it's standard to present MDS (or PCA) results in that manner, and including labels could potentially overload the plot. We hope the reviewer agrees.

Fig S6: As mentioned above, it is not clear how these pseudo-coordinates should be interpreted. Maybe clarify this in the caption?

We have now provided a quantification of the euclidean distances between the geographical coordinates and pseudo-coordinates (based on musical instruments, subsistence tools and genes) of the 10 CAHGs (See Table S10). We can now estimate correlations and regressions between them and properly quantify their possible 'match'. For example, the distance between the geographic locations and pseudo-coordinates estimated from musical instruments was generally smaller (for 7/9 CAHG groups) than those between the geographical locations and pseudo-coordinates estimated from subsistence tools ($W=20$; $P=0.038$) (see Figure below, also included as Fig S18). This means that the structure of variation in musical instruments mapped better onto the geographical distribution of groups. We have added such a statement to the main text.

Fig S7-9: none of them are referred in the text

We have now referred to all supplementary tables and figures in the text.

Fig S10 and S13: According to these figures, there are 10 groups with musical instruments but 11 with names for them, as well as 10 groups with subsistence tools but only 9 with names for them. Please clarify what this means (are there subsistence tools attested in one community but no name for them, and names for musical instruments in one community where the instrument is not attested?) Also, musical instruments correspond to the top plot in Fig S13 and to the bottom plot in Fig S10, which can be confusing.

For clarity, we have now removed the Mbendjele from the linguistic analyses so the groups included in both linguistic and material culture analyses are the same (see Tables S1-S3). There are 9 groups for which we have subsistence tool terminology available simply because we had no records of specialised vocabulary for subsistence technology for the Bedzan. We hope the new version of Fig. S17 and Materials and Methods section make this clearer.

Fig S12: how much of the variance is explained by each principal coordinate?

We have now included Fig S3 showing the percentage of variance and cumulative percentage of variance explained by each principal coordinate for musical instrument, subsistence tools and CAHG genetic components (to match the heatmaps in Fig. S2),

Fig S16: labels are in many cases hidden by the arrows, please jitter the former to increase readability.

The *SpaceMix* algorithm places the labels in the location of the geographical coordinates of the actual population (bold letters) and the inferred admixture source (italics). Hence, the labels carry meaningful spatial information and thus cannot be moved. This information is included on the caption to the figure (now Fig. S13).

Fig S17: Does this distribution imply that for a Masked $F_{st} = 0$ (identical populations) the predicted number of shared musical instruments is zero / very low, and the maximum is only at Masked $F_{st} = 0.01$?

We agree that this Figure was confusing. It referred to the number of words shared between groups, as opposed to the probability of sharing vs. not sharing any words. We have for this reason removed this figure and left the results from all the Zero-Inflated Poisson models in Tables S14-S19 to prevent confusion.

Table S1: In the first paragraph it is stated that 14 languages are spoken by these 11 CAHG groups, and Fig 1 and this table are referred to, but no information of languages appears in any of them. Maybe add the language information to this table?. Also, what are the units for Area?

We have now modified the reference and indeed pointed readers to Tables S2-S3 where all the linguistic information is included. The area is in squared degrees, and we have now modified Table S1 to indicate this.

Table S4: missing according to the current numeration

We have relabelled and re-organized all the SM, and now there are no inconsistencies in the numbering.

Design: spaces between tables starting at Table S10-11 are shifted, leaving the captions and sometimes the first row on one page and the rest of the table in the other

This is not the case in our computer, but we will reformat to make sure it doesn't happen.

Tables S12-13: see above (language classification).

These tables have now been updated. Please see our answer to the reviewer's major comment number 2.

Table S14: last column changes the order in which the populations are presented in different rows, e.g. "Aka, Baka" in one row and "Baka, Aka" in another. Maybe homogenize it with alphabetical order or another consistent order?

We have modified the table (now Table S13) accordingly, and ordered groups alphabetically.

Table S15: duplicated name for 2 different tables

This has now been corrected.

Table S15 (first one): It is not clear to which languages each of the abbreviations refer. These are neither the ones defined in Table S12 or in Table S13 (see table in Major comments).

These tables have now been updated. Please see our answer to the reviewer's major comment number 2.

Finally, we would like to thank the reviewer very much for all the insightful and careful comments. We want to say that the comments above have truly guided us in producing an improved version of our manuscript. We hope the reviewer is satisfied with the modifications and justifications of our procedures. Thanks again.

Reviewer #3:

Remarks to the Author:

This innovative and important study by Padilla-Iglesias et al makes a significant contribution to understanding the drivers of cultural evolution in Central Africa. More generally it presents a case study illustrating the mechanisms of cultural and social evolution, as well as providing a methodological template that will inspire future studies. It is a perfect fit for Nature Human Behaviour and I would be delighted to see it published as soon as possible.

The 'new' data consists of a very significant inventory of nearly 1400 musical items and subsistence tools. This has been carefully gathered from a range of sources. This is supplemented with the identification of shared lexical items between the groups under study. Genetic data have been gathered from material already publicly available. The between-group variation in these are compared with geography and environment.

The paper is very innovative in the way it compares genomic and cultural indicators of network distance as well as geography and linguistics. Statistical operations have been done appropriately, including correcting for multiple comparisons where appropriate.

Genomic analysis is very well explained and seems well executed with the masking of segments from more recent mixing irrelevant to what is being studied. This has led to the fascinating conclusion that

musical instruments reflect a deep historical connectivity between groups that contrasts with the more geographically-determined adaptations seen in subsistence tools.

The paper is extremely well written and organised. Figures generally excellent, Figure 4 is a particularly good at conveying how shared material culture can be used to reconstruct historical connections between societies over considerable distances.

I have only a few suggestions for improvement:

There are a few miss-numbered references and cross-references to the tables in the SI.

We thank the reviewer very much for the comments. This has now been corrected.

In the analysis of material culture the authors have used presence / absence data to calculate Jaccard distances for cultural items. it was not entirely clear from the main part of the paper whether these are founded on the presence / absence of instruments or tools, or the presence / absence of distinctive traits found on the objects themselves.

We have modified the text to specify that Jaccard distances were calculated from the presence/absence of musical instrument types (see the first results subsection “*Musical instruments of CAHGs result from geographically extended cultural exchange*”). The same was done for presence or absence of foraging tools.

Looking at the supplementary information, this becomes clear. However, without access to the actual contingency tables or computer code used in the analysis, it is not possible to access whether simple errors in the coding present in the S1 table (e.g axe identified as ‘ax’, ‘axe’ and ‘Axe’; there are other examples) influence the statistical results.

As also noted by review 2, we should have made the repository available for the review process. We are rectifying this issue now. Upon acceptance of the manuscript, it will be made available for the general public as well, but in the meantime, reviewers have special means to access the data and code in the

following link:

<https://www.dropbox.com/scl/fo/gj4qg3wp73jfovpm9hkn3/h?rlkey=uml9a3js1drmcwec6fkwxrwqu&dl=0>

We have indicated this instructions in the manuscript file as well

As for Dataset S1, it was not automatically processed for analyses. Another manually-made table with presence/absence of each item each population was used instead in the analyses (please see the repository for a copy). Hence, despite slight variation in the labelling of items in Dataset S1, any population having for example “arrows, Arrow, Arrows etc..” was coded as having arrows present. However, we have nonetheless updated Dataset S1 for consistency according to the reviewer’s suggestion.

On ecological distance and biome type – the authors used the ‘dataset’ provided by Beyer et al 2020. This is not a dataset but rather a global vegetation model (which is based on BIOME4 equilibrium global vegetation model first used in experiments described in Kaplan et al. (2003)). The dissimilarities used to reconstruct the ecological ‘distance’ were therefore seemingly based on the number of different biomes contained within the territory of each group. I’d worry that this is merely reflecting the size of territory. It would be better to find some way of understanding what the real ecological ‘distance’ is between the various zones in the BIOME4 model.

Thank you very much for this remark. We have changed the word “dataset” for “global vegetation model” in the text. The ecological distance included in the main text is not based on the number of biomes contained in each territory, but on the percentage of the territory occupied by particular biomes using Gower dissimilarity distances. In this way, a large territory where 95% is covered by one biome and the remaining 5% by 5 different ones and another one covered 100% by the same biome will have a similar proportion of the first biome and thus be deemed similar. For example, in our study, the Baka (group with the largest surface area) had 71% of its territory covered by tropical evergreen forest, 21% by tropical semi-deciduous forest and 6% by tropical savannah. The Bakola and Bakoya’s territory is 100% covered by tropical evergreen forest (and they are some of the groups with the smallest territory), hence, they have the same overarching biome but the other two present in the Baka’s territory are absent. In contrast, the territory from the Aka (the second largest) is covered 37% by tropical evergreen forest, 54% by tropical semi-deciduous forest and 12% by tropical savannah. Hence, it has the same biome types as the Baka, but not in the same proportions. Nonetheless, the ecological distance between the Baka and the Aka (0.12) Bakola (0.11) and Bakoya (0.11) are fairly similar. By contrast, the Eastern Batwa, for which almost half (47%) of their territory is covered in tropical xerophytic shrubland, have a much higher ecological distance with the Baka (0.63) even if there is also a small proportion of tropical semi-deciduous forest

(21%), tropical evergreen forest (5%), tropical deciduous forest (11%) and temperate conifer forest (16%). Please see Supplementary figures 10 and 11 for details.

We also ran the analyses using ecological distances based on the presence/absence of biome types and the results were not qualitatively different to those reported in the main paper. For this reason, we decided not to include those results in the SM (although the code to extract this information can be found in the link provided). However we would like to reinstate that our approach avoids a potential effect of territory size by being based on proportions.

We would like to thank the reviewer again for the supportive comments and very useful notes that assisted us in revising the manuscript. We hope the reviewer is satisfied with our answers and modified manuscript.

Reviewer #4:

Remarks to the Author:

The study by Padilla-Iglesias analysed ethnographic, linguistic and genetic data of Central African rainforest hunter-gatherer (RHG) populations. Genetic data were collected from previous genome-wide studies of RHGs and then analysed together with cultural data (music instruments, subsistence tools and linguistic data) associated with them to infer their demographic history and to correlate cultural and genetic exchange between these African groups. The study seems to be a comprehensive study of African RHG, however the manuscript needs clarifications about the genetic methods and the analysed datasets. I included detailed comments below to improve the clarity and text in the manuscript.

Major comments:

Figure 1 is lacking clarity. The authors analysed different datasets (music instruments, subsistence tools, linguistic, and genetic data), several results are presented for each dataset with different names for the populations/groups, and some populations are missing or have different names in the genetic dataset. It could be easier to understand the analysed populations/groups with two or more maps with the location of the populations/groups included in each analysed dataset.

We agree that figures had some inconsistencies. We have relabelled all supplementary tables and figures so that the labels of the populations match those in Figure 1 and re-done Figure 1 for clarity.

Also, please see Table S4, where we explain the sample size of each group and which analyses they were included in.

It's also confusing that the name of the populations included in the genetic dataset does not match the names of the populations in the previous studies, making it more difficult to compare the results in Fig S1-S5 with results presented in previous studies (Patin et al. 2014; 2017; Perry et al. 2014).

We understand the issue. This points to a broader problem of disciplinary boundaries, as linguists, anthropologists and geneticists haven't used the same names to define populations. We have now added an extra column in Table S4 indicating (for clarity) the name(s) used for each group in the original datasets/publications they belonged to.

As the authors and previous studies pointed out, there are genetic differences between Western and Eastern RHG. For the local ancestry de-convolution, the authors could explain the motivation to focus only on one "CAHG ancestry component" when they could also investigate the Western and Eastern ancestry components. In addition, in the section methods (page 19), the authors could indicate the sample size and samples included in the reference panel for each ancestry.

Thank you for this suggestion. We have now indicated in our Methods section the size of the reference panel for each ancestry. Nonetheless, the Gnomix compensates for imbalance referenced panels with the *r_admixed* parameter which we set to 1, so that the programme simulated an even number of reference and admixed individuals when training the model.

Regarding Eastern and Western CAHG components, we did report them in Table S21 but since the focus of the paper was on assessing genetic and cultural connectivity across the Congo Basin, we combined Eastern and Western CAHG components.

For comparisons between datasets and Mantel test results (Table 1), the authors could include heatmap plots for genetic distances (e.g., Fst for populations in each dataset), another heatmap plot for geographical distances (e.g., Km), and another heatmap plot for linguistic distances (e.g., PMI-weighted distance?).

Thank you very much for this suggestion. We had already included F_{st} heatmap plots so that readers could compare between-population F_{st} values at each stage of the multi-stage genomic masking process we used (See Figs S7-S9). We also had included heatmaps of the Jaccard distances between populations based on shared musical instruments and subsistence tools, which we now have combined with heatmaps of ecological distance and geographic distance as suggested by the reviewer (Fig. S11).

In addition, we have made another panel plot (Fig. S16) with the PMI-weighted linguistic distances and the patristic distances newly calculated as suggested by another reviewer, so that readers can compare values obtained from both methods. We did not create an equivalent heatmap for the Glottolog based linguistic distances, as they simply represent the count of the number of nodes in the Glottolog tree, which we have included in Fig. S15).

The Discussion section should discuss more the genetic results presented in this study and put the new results in the context of previous studies. So far this section seems to discuss only the cultural results.

Thanks for the suggestion. We have added a few sentences to the Discussion to emphasise the results obtained with our proposed analysis of IBDs in three stages.

The manuscript could also comment on the exchange of musical instruments between Bantu-speaking populations and RHG, to explain the high diversity of instruments in Figure 4A in comparison with the subsistence tools in Figure 4B.

We have now added a SM table (S13), showing which musical instrument words (as well as musical instruments) are shared between CAHGs and farmers. They are a small proportion of the total (7 out of 23). Although for most of these cases the directionality of sharing is reported in ethnographies to be from CAHG to farmers, we have now excluded these cases from Figure 4A (which represents the sharing of musical instrument words, as opposed to musical instruments themselves), in order to focus on the exchange between CAHGs exclusive of farmers, and remain conservative. We have also included a column in Dataset S2 containing additional information regarding the geographic distribution of each of the words, and available information regarding its origins.

Minor comments:

Table S15, were all the studied populations included in the ASJP dataset? This supplementary table doesn't include all the RHG and a few names of Bantu-speaking populations. Could the authors better indicate the ASJP code of each studied population in the linguistic dataset? A heatmap plot of linguistic data could also help to understand the linguistic diversity of the studied populations.

Yes, all the languages from the studied populations (or of a linguistically close neighbour) were included in the ASJP dataset. The codes of the languages ISO-639-3 codes, Glottocodes and Ethnologue classifications of all the languages included can now be found in Table S3. As stated above, we have now included heatmaps showing the genetic diversity of the populations included in the study.

For ADMIXTURE analyses. How many iterations were run at each K-group and whether a random seed was used each time?

A random seed was used for each K value. We have now specified this in the text. The number of iterations run for each K value can be seen in the table below, which we have now included as a supplementary table (Table S20). Therefore, for each value of K, we ran as many iterations as required for the log-likelihood to increase by less than $\epsilon=10^{-4}$ between iterations.

K value	Iterations
2	58
3	53
4	50
5	46
6	55
7	61
8	50

The authors performed ADMIXTURE analyses from K=2 to K=8, however there are only suppl figures from K=2 to K=5. Other K0-groups could help to understand the population structure in Central Africa.

We have added Figures S20 and S21 now with K=2 to K = 8.

For local ancestry inference and masking, did the authors use any threshold to avoid miss-classify haplotypes?

Since *Gnomix* is a machine learning method that relies on training an algorithm on a subset of the data and testing it on another subset of the data, rather than imposing a threshold it estimates the classification accuracy for each ancestry by producing for each chromosome a confusion matrix reporting how many of the test haplotypes are correctly classified and miss-classified per ancestry group.

We have now added this confusion matrix to the Supplementary Material (Table S22), and reported in the text that on average, across chromosomes, the classification accuracy for haplotypes of CAHG ancestry was of 88% (range= 83-95%) and for haplotypes of agriculturalist ancestry of 84% (range=76-88%). This is an extremely good performance for a local ancestry inference method, especially on SNP array data and given the time depth of admixture between CAHG and Bantu. For example, if we assume a date of admixture around 80 generations ago, the classification accuracy of *RFMix* (the most widely used local ancestry inference method) drops to less than 70% (see Hilmarsson et al. 2021 for more details on how *Gnomix* outperforms other available softwares even when data at a much higher resolution is used in other softwares).

For the phasing procedure, the authors could indicate whether they used reference panels and recombination maps. If that's the case, information about them is relevant.

We used the GRCh37 genetic map (as indicated in the Materials and Methods section) but we didn't use a reference panel. This is a standard procedure when processing human genomic data for masking (and many other) purposes. See Ioannidis et al. 2020 *Nature* and 2021 *Nature* or Schlebusch et al. 2020 *Molecular Biology and Evolution*, or Arango-Isaza et al. 2023 *Current Biology* for recent publications where the same processing was used for masking purposes.

In Fig S4 legend, the text indicated those are MDS results, however in the plot the x-axis and y-axis are indicating: PCA1 and PCA2.

Thank you for noticing that. Sorry, the legend was wrong - although we performed both types of analyses, the ones reported in the paper are indeed MDS (for comparability across datasets). We have corrected this accordingly.

Discussion: change culture data for anthropological or ethnographic data, because linguistics is part of the culture.

We have specified now “material cultural”, to contrast it with lexical data.

We would like to thank the reviewer for the very careful read of our manuscript. The comments were very useful, and we believe that their incorporation resulted in a significantly improved manuscript.

Decision Letter, first revision:

21st December 2023

Dear Ms Padilla-Iglesias,

Thank you once again for your manuscript, entitled "Deep history of cultural and linguistic evolution among Central African hunter-gatherers," and for your patience during the peer review process.

Your manuscript has now been evaluated by the 4 reviewers from the previous round, whose comments are included at the end of this letter. As you will see, three of our reviewers are satisfied with your revisions pending some minor concerns; however, Reviewer 2 raises a concern that could impact your conclusions. We are very interested in the possibility of publishing your study in Nature Human Behaviour, but would like to consider your response to Reviewer 2's concerns in the form of a revised manuscript before we make a decision on publication. Please also address the minor points raised by our other referees.

Finally, your revised manuscript must comply fully with our editorial policies and formatting requirements. Failure to do so will result in your manuscript being returned to you, which will delay its consideration. To assist you in this process, I have attached a checklist that lists all of our

requirements. If you have any questions about any of our policies or formatting, please don't hesitate to contact me.

In sum, we invite you to revise your manuscript taking into account all reviewer and editor comments. We are committed to providing a fair and constructive peer-review process. Do not hesitate to contact us if there are specific requests from the reviewers that you believe are technically impossible or unlikely to yield a meaningful outcome.

We hope to receive your revised manuscript within two months. I would be grateful if you could contact us as soon as possible if you foresee difficulties with meeting this target resubmission date.

- Include a "Response to the editors and reviewers" document detailing, point-by-point, how you addressed each editor and referee comment. If no action was taken to address a point, you must provide a compelling argument. When formatting this document, please respond to each reviewer comment individually, including the full text of the reviewer comment verbatim followed by your response to the individual point. This response will be used by the editors to evaluate your revision and sent back to the reviewers along with the revised manuscript.
- Highlight all changes made to your manuscript or provide us with a version that tracks changes.

[REDACTED]

We look forward to seeing the revised manuscript and thank you for the opportunity to review your work. Please do not hesitate to contact me if you have any questions or would like to discuss these revisions further.

Sincerely,

[REDACTED]

Senior Editor
Nature Human Behaviour

REVIEWER COMMENTS:

Reviewer #1:

Remarks to the Author:

The revised manuscript reads much clearer and the revised descriptions of the methods (particularly IBD) makes more sense to me than in the original version. Thank you for a detailed written rebuttal (although you completely misunderstood my point about Hodder's Baringo insights, you nevertheless included better ethnographic insights to justify your results).

I still request that a legend be added to Fig. 1 to clarify the colour scheme. Note that red-green colourblind individuals will not be able to read this figure at all, so I would recommend a simple change to the colour scheme and a legend added to the map on what your three colours denote. The figure caption tells us what dark green and red represent but not light green. Sorry to put you through the hoops, but I did test this with a colourblind individual personally close to me and they confirmed that it is unreadable.

Some of the references need to be fixed as well: 8, 15, 17, 18, 20, etc... The authors need to include full citation information, including dois, where possible.

That said, I agree with the authors that the revised approach of the manuscript does represent a new methodological tack, provides interesting insights into the migration histories of Central Africa and conservation and adoption of certain terms across space and time. This paper no longer reads as a 'forest of methods', but now has a clear narrative structure and point that is accessible to non-(genetic/linguistic) specialists like me who are interested in Central African demography and culture histories.

Reviewer #2:

Remarks to the Author:

I'm happy to see that many changes suggested by the different reviewers have been made in this new version of the manuscript, significantly improving its quality and readability. In particular, the detailed review of linguistic data, consulting expert linguists and grammars of the languages involved, make the linguistic claims much more robust.

I only have one concern, with respect to the linguistic distances claimed to not be associated with the specialized lexicon of musical instruments, while they indeed are, according to the results shown (first comment). I wonder how this affects the conclusions. Besides this, I have several minor and very minor comments below it.

Main comment:

Line 261

"Zero Inflated Poisson models revealed that genetic and geographic distance (but neither PMI, Glottolog, or patristic distances) were significant predictors of the number of shared musical instrument words between populations (Tables S14-S19). This association between shared musical instrument lexicon with deep population history, but not with overall linguistic relationships or similarities, held both when considering all shared terms between CAHGs, and when only considering

those unique to CAHG.”

This is not correct, as from the results in Tables S14-19. In particular, in Tables S16 and S19, where patristic distance is used (the most accurate in my opinion among the 3 linguistic distances considered), significant results are found both in the model including Genetic distance (CAHG ancestry) and the one including Geographical distance in Table S16 (full vocabulary), and it is significant for the model involving Geographical distance in Table S19 (vocabulary unique to CAHGs). Please comment on this fact, and whether considering these changes anything in the conclusions and description of results.

Minor comments (in order of appearance in the main text):

Table S5

Ecological distance appears in this table, cited on line 116, but the definition of this distance doesn't appear until line 134. Maybe referring to Fig S10 here would help.

Table S9

I would sort this table by value of Proportion of genome from inferred admixture source (there is no clear sorting of the CAHGs in the current form)

Table S2

Languages are described by Phylum, Stock, Family, Group, and Sub-group, but the languages themselves are not mentioned.

Table S3

Ethnologue classification is sometimes underlined and sometimes not. Edit consistently.

Fig S16

PMI linguistic distance hasn't been defined in the text before. Also, clarify in the caption (in addition to the figure title) which distance is in the left and which in the right.

Table S14-19

Just stylistic: For clarity and consistency with previous tables, mark in bold the whole row of significant results, not only the p-value.

Lines 305-313

Maybe compare words for subsistence tools with patristic distances just like it was done with musical instruments? This could help disentangle whether they were borrowed from farmers' languages.

Lines 370-374

“By contrast, the fact that very few tool-related words are shared between CAHGs may reflect either a convergent origin of the similar tools in groups with divergent languages (and hence distinct words created for similar tools); or an ancient common ancestor to both subsistence tools and tool-related words, with tool (but not word) differentiation over time being limited by ecological constraints.” The second option looks to me straightforward to check: just compare the cases of the same tool existing in different groups: whether or not its word form also coincides. Does this make sense?

Lines 585 to 634

This subsection "Assessing the sharing of terms to design musical instruments and subsistence tools", seems to be duplicated, with only the first sentence changed significantly.

Line 651

"For the Babongo, the only linguistically heterogeneous population in our sample, we averaged pairwise distances based on each of its languages."

According to Tables S3, Batwa (East) speaks 2 different languages, making it linguistically heterogeneous as well. In addition, Table S3 shows Babongo speaking 3 languages and Batwa (East) 2, while Table 4 shows Babongo speaking 4 languages and Batwa (East) only 1.

Reviewer #3:

Remarks to the Author:

My only significant concern about this paper when I first reviewed it concerned the derivation of ecological distance from a vegetation model. But thanks to the authors' very clear explanation of how they have calculated and dealt with the Gower similarity, I am satisfied that their analysis is appropriate and supports the conclusions in the paper. I have looked at the repository of data and code now available, which makes it clear that this is a rigorous and reproducible analysis of the data. Other aspects of the paper have been corrected and clarified too, and in summary I think constitutes an important contribution to the study of culture through deep time. Rowan McLaughlin.

Reviewer #4:

Remarks to the Author:

The authors thoroughly addressed all my comments and remarks, and the manuscript has notably improved as a result. I therefore recommend this manuscript for publication.

Author Rebuttal, first revision:

REVIEWER COMMENTS:

Reviewer #1:

Remarks to the Author:

The revised manuscript reads much clearer and the revised descriptions of the methods (particularly IBD) makes more sense to me than in the original version. Thank you for a detailed written rebuttal (although you completely misunderstood my point about Hodder's Baringo insights, you nevertheless included better ethnographic insights to justify your results).

I still request that a legend be added to Fig. 1 to clarify the colour scheme. Note that red-green

colourblind individuals will not be able to read this figure at all, so I would recommend a simple change to the colour scheme and a legend added to the map on what your three colours denote. The figure caption tells us what dark green and red represent but not light green. Sorry to put you through the hoops, but I did test this with a colourblind individual personally close to me and they confirmed that it is unreadable.

We thank very much the reviewer for his positive feedback and we are sorry that our figure was unreadable for colour blind individuals. We have now modified the red for light pink, added the writing in black (to increase contrast) and added a colour legend.

Some of the references need to be fixed as well: 8, 15, 17, 18, 20, etc... The authors need to include full citation information, including dois, where possible.

Thank you very much for this remark. We have now fixed all the references (included the pinpointed ones).

That said, I agree with the authors that the revised approach of the manuscript does represent a new methodological tack, provides interesting insights into the migration histories of Central Africa and conservation and adoption of certain terms across space and time. This paper no longer reads as a 'forest of methods', but now has a clear narrative structure and point that is accessible to non-(genetic/linguistic) specialists like me who are interested in Central African demography and culture histories.

Again, we thank the reviewer for the positive feedback and for previous suggestions on how to improve our manuscript.

Reviewer #2:

Remarks to the Author:

I'm happy to see that many changes suggested by the different reviewers have been made in this new version of the manuscript, significantly improving its quality and readability. In particular, the detailed review of linguistic data, consulting expert linguists and grammars of the languages involved, make the linguistic claims much more robust.

I only have one concern, with respect to the linguistic distances claimed to not be associated with the specialized lexicon of musical instruments, while they indeed are, according to the results shown (first comment). I wonder how this affects the conclusions. Besides this, I have several minor and very minor comments below it.

Main comment:

Line 261

“Zero Inflated Poisson models revealed that genetic and geographic distance (but neither PMI, Glottolog, or patristic distances) were significant predictors of the number of shared musical instrument words between populations (Tables S14–S19). This association between shared musical instrument lexicon with deep population history, but not with overall linguistic relationships or similarities, held both when considering all shared terms between CAHGs, and when only considering those unique to CAHG.”

This is not correct, as from the results in Tables S14–19. In particular, in Tables S16 and S19, where patristic distance is used (the most accurate in my opinion among the 3 linguistic distances considered), significant results are found both in the model including Genetic distance (CAHG ancestry) and the one including Geographical distance in Table S16 (full vocabulary), and it is significant for the model involving Geographical distance in Table S19 (vocabulary unique to CAHGs). Please comment on this fact, and whether considering these changes anything in the conclusions and description of results.

We thank very much the reviewer for this remark and apologise for the lack of clarity. The main point of the analyses relating phylogenetic linguistic distances and the sharing of musical instrument vocabulary is to assess whether groups that are closer linguistically are more likely to share musical instrument terminology. Our hypothesis is that the sharing of musical instrument terminology is *not due* to phylogenetic linguistic proximity but due to these terms being older than the adoption of farmer languages by CAHG. Our results support this hypothesis as in fact, the significant association the reviewer points out to from Table S16 goes in the opposite direction. That is, it tells us that groups speaking the most distantly related languages are more likely to share musical instrument terminology. We have clarified this in the main text (lines 264–269), which now writes as follows:

Zero Inflated Poisson models revealed that genetic and geographic proximity were significant predictors of the number of shared musical instrument words between populations, whilst phylogenetic linguistic proximity and overall language similarity (i.e. smaller patristic, PMI or Glottolog distances) were not (Tables S14–S19). In fact, when considering patristic distances, we found the opposite to be true, with groups most distantly related linguistically sharing more share musical instrument words (Table S16, S19).

Minor comments (in order of appearance in the main text):

Table S5

Ecological distance appears in this table, cited on line 116, but the definition of this distance doesn't appear until line 134. Maybe referring to Fig S10 here would help.

In that sentence (line 116) we are not referring yet to any results regarding biomes. On line 116 we only refer to the association between geographic distance and musical instrument distance.

Table S9

I would sort this table by value of Proportion of genome from inferred admixture source (there is no clear sorting of the CAHGs in the current form)

Thank you very much for this suggestion. We have modified the table accordingly.

Table S2

Languages are described by Phylum, Stock, Family, Group, and Sub-group, but the languages themselves are not mentioned.

Table S2 is based on the linguistic classification of the CAHG languages from Bahuchet (2006; 2012), where he places the languages spoken by each group within their respective phylogenies without specifying a name for them (only specifying the closest farmer language to them, which we have also specified in Table S3).

Table S3

Ethnologue classification is sometimes underlined and sometimes not. Edit consistently.

Thank you very much for the careful read. We have edited the table accordingly.

Fig S16

PMI linguistic distance hasn't been defined in the text before. Also, clarify in the caption (in addition to the figure title) which distance is in the left and which in the right.

We have edited the legend of the figure to include which distance is on the left and which one on the right, as well as included *Pointwise Mutual Information Distance* (instead of the acronym), and pointed readers to the Methods section for further information. The first reference to this figure in line 212 is simply to illustrate to the readers that different groups speak differentiated languages. Then, in the second reference to that figure in line 252, a description of the measure is provided.

Table S14-19

Just stylistic: For clarity and consistency with previous tables, mark in bold the whole row of significant results, not only the p-value.

We have modified the tables accordingly.

Lines 305-313

Maybe compare words for subsistence tools with patristic distances just like it was done with musical instruments? This could help disentangle whether they were borrowed from farmers' languages.

As specified in lines 319-321, the very few terms shared between CAHG groups prevented us from performing similar statistical assessments linking patristic distances and the sharing of subsistence tool terminology as we did with musical instrument terminology (and of particular importance, only 3 words were exclusive to CAHG groups).

Lines 370-374

“By contrast, the fact that very few tool-related words are shared between CAHGs may reflect either a convergent origin of the similar tools in groups with divergent languages (and hence distinct words created for similar tools); or an ancient common ancestor to both subsistence tools and tool-related words, with tool (but not word) differentiation over time being limited by ecological constraints.”

The second option looks to me straightforward to check: just compare the cases of the same tool existing in different groups: whether or not its word form also coincides. Does this make sense?

We understand the reviewer's point, and we agree that the second alternative is more parsimonious. However, while we provided the list of very few tools whose names are shared among groups, for the vast majority of shared tools (which are many, see the *P_A_subsistence.csv* file in the data folder), the words used to designate them do not coincide.

Thus, although we tend to agree with the reviewer, we believe the current data do not allow us to differentiate between our two proposed alternatives.

Lines 585 to 634

This subsection “Assessing the sharing of terms to design musical instruments and subsistence tools”, seems to be duplicated, with only the first sentence changed significantly.

Thank you so much for the careful read. This was a problem from the “Track changes” function in Microsoft Word. We have removed the duplicated section.

Line 651

“For the Babongo, the only linguistically heterogeneous population in our sample, we averaged pairwise distances based on each of its languages.”

According to Tables S3, Batwa (East) speaks 2 different languages, making it linguistically heterogeneous as well. In addition, Table S3 shows Babongo speaking 3 languages and Batwa (East) 2, while Table 4 shows Babongo speaking 4 languages and Batwa (East) only 1.

We think there is a misunderstanding here as table S3 only shows the Batwa (East) speaking one language.

Reviewer #3:

Remarks to the Author:

My only significant concern about this paper when I first reviewed it concerned the derivation of ecological distance from a vegetation model. But thanks to the authors' very clear explanation of how they have calculated and dealt with the Gower similarity, I am satisfied that their analysis is appropriate and supports the conclusions in the paper. I have looked at the repository of data and code now available, which makes it clear that this is a rigorous and reproducible analysis of the data. Other aspects of the paper have been corrected and clarified too, and in summary I think constitutes an important contribution to the study of culture through deep time. Rowan McLaughlin.

We thank the reviewer for their positive feedback.

Reviewer #4:

Remarks to the Author:

The authors thoroughly addressed all my comments and remarks, and the manuscript has notably improved as a result. I therefore recommend this manuscript for publication.

We thank very much the reviewer for their positive feedback.

Decision Letter, second revision:

11th March 2024

Dear Dr. Padilla-Iglesias,

Thank you for your patience as we've prepared the guidelines for final submission of your Nature Human Behaviour manuscript, "Deep history of cultural and linguistic evolution among Central African hunter-gatherers" (NATHUMBEHAV-23030740B). Please carefully follow the step-by-step instructions provided in the attached file, and add a response in each row of the table to indicate the changes that you have made. Please also check and comment on any additional marked-up edits we have proposed within the text. Ensuring that each point is addressed will help to ensure that your revised manuscript can be swiftly handed over to our production team.

We would hope to receive your revised paper, with all of the requested files and forms within two-three weeks. Please get in contact with us if you anticipate delays.

Nature Human Behaviour offers a Transparent Peer Review option for new original research manuscripts submitted after December 1st, 2019. As part of this initiative, we encourage our authors to support increased transparency into the peer review process by agreeing to have the reviewer comments, author rebuttal letters, and editorial decision letters published as a Supplementary item. When you submit your final files please clearly state in your cover letter whether or not you would like to participate in this initiative. Please note that failure to state your preference will result in delays in accepting your manuscript for publication.

In recognition of the time and expertise our reviewers provide to Nature Human Behaviour's editorial process, we would like to formally acknowledge their contribution to the external peer review of your manuscript entitled "Deep history of cultural and linguistic evolution among Central African hunter-gatherers". For those reviewers who give their assent, we will be publishing their names alongside the published article.

Cover suggestions

We welcome submissions of artwork for consideration for our cover. For more information, please see our guide for cover artwork.

ORCID

Non-corresponding authors do not have to link their ORCIDs but are encouraged to do so. Please note that it will not be possible to add/modify ORCIDs at proof. Thus, please let your co-authors know that if they wish to have their ORCID added to the paper they must follow the procedure described in the following link prior to acceptance:

Nature Human Behaviour has now transitioned to a unified Rights Collection system which will allow our Author Services team to quickly and easily collect the rights and permissions required to publish your work. Approximately 10 days after your paper is formally accepted, you will receive an email in providing you with a link to complete the grant of rights. If your paper is eligible for Open Access, our Author Services team will also be in touch regarding any additional information that may be required to arrange payment for your article.

Please note that *Nature Human Behaviour* is a Transformative Journal (TJ). Authors may publish their research with us through the traditional subscription access route or make their paper immediately open access through payment of an article-processing charge (APC). Authors will not be required to make a final decision about access to their article until it has been accepted. Find out more about Transformative Journals

[REDACTED]

Best regards,

[REDACTED]

Editorial Assistant

Nature Human Behaviour

On behalf of

[REDACTED]

Senior Editor

Nature Human Behaviour

Reviewer #2:

Remarks to the Author:

I am satisfied with the answers to all of my comments, and I believe the article is ready for publication from my end. I am just writing down a few very minor comments, just for the consideration of the authors, for which I don't need any further answer, given the little impact they might have in the final version.

1. About my main comment:

I agree with the authors in that their conclusions hold, for the correlation with patristic distances is in the direction opposite to that expected, as explained by the authors.

A minimal writing comment: "with groups most distantly related linguistically sharing more share musical instrument words" → "sharing more musical instrument words" OR "having more shared musical instrument words"

2.

>> Table S3

>> Ethnologue classification is sometimes underlined and sometimes not. Edit consistently.

>>> Thank you very much for the careful read. We have edited the table accordingly.

Baka* and Aka are still not underlined (sorry for being this annoying!)

3.

>> Line 651

>> "For the Babongo, the only linguistically heterogeneous population in our sample, we averaged pairwise distances based on each of its languages."

According to Tables S3, Batwa (East) speaks 2 different languages, making it linguistically heterogeneous as well. In addition, Table S3 shows Babongo speaking 3 languages and Batwa (East) 2, while Table 4 shows Babongo speaking 4 languages and Batwa (East) only 1.

>>> We think there is a misunderstanding here as table S3 only shows the Batwa (East) speaking one language.

You are right: there was a mistake in my table labels. What I meant was:

According to Table S2, Batwa (East) speaks 2 different languages (language sub-groups J11 and J60), making it linguistically heterogeneous as well. In addition, Table S2 shows Babongo speaking 3 languages (language sub-groups B30, B60, and B70) and Batwa (East) speaking 2, while Table S3 shows Babongo speaking 4 languages (Mbeté (B602), Teke (B75), Tsogo (B31), and Nzebi (B52)) and Batwa (East) only 1 (Ruanda-Rundi (D61)).

However, I understand that the authors are using the classification in Table S3 for calculating linguistic distances. Therefore, the only linguistically heterogeneous group according to this classification is indeed Babongo, so no changes are needed.

Author Rebuttal, second revision:

Reviewer #2:

Remarks to the Author:

I am satisfied with the answers to all of my comments, and I believe the article is ready for publication from my end. I am just writing down a few very minor comments, just for the consideration of the authors, for which I don't need any further answer, given the little impact they might have in the final version.

1. About my main comment:

I agree with the authors in that their conclusions hold, for the correlation with patristic distances is in the direction opposite to that expected, as explained by the authors.

A minimal writing comment: "with groups most distantly related linguistically sharing more share musical instrument words" → "sharing more musical instrument words" OR "having more shared musical instrument words"

Thank you very much for the feedback and the careful read. We have now corrected this in line 267 of the manuscript.

2.

>> Table S3

>> Ethnologue classification is sometimes underlined and sometimes not. Edit consistently.

We have now removed the underlining.

>>>> Thank you very much for the careful read. We have edited the table accordingly.

Baka* and Aka are still not underlined (sorry for being this annoying!)

We have now removed the underlining.

3.

>> Line 651

>> “For the Babongo, the only linguistically heterogeneous population in our sample, we averaged pairwise distances based on each of its languages.”

According to Tables S3, Batwa (East) speaks 2 different languages, making it linguistically heterogeneous as well. In addition, Table S3 shows Babongo speaking 3 languages and Batwa (East) 2, while Table 4 shows Babongo speaking 4 languages and Batwa (East) only 1.

>>>> We think there is a misunderstanding here as table S3 only shows the Batwa (East) speaking one language.

You are right: there was a mistake in my table labels. What I meant was:

According to Table S2, Batwa (East) speaks 2 different languages (language sub-groups J11 and J60), making it linguistically heterogeneous as well. In addition, Table S2 shows Babongo speaking 3 languages (language sub-groups B30, B60, and B70) and Batwa (East) speaking 2, while Table S3 shows Babongo speaking 4 languages (Mbeté (B602), Teke (B75), Tsogo (B31), and Nzebi (B52)) and Batwa (East) only 1 (Ruanda-Rundi (D61)).

However, I understand that the authors are using the classification in Table S3 for calculating linguistic distances. Therefore, the only linguistically heterogeneous group according to this classification is indeed Babongo, so no changes are needed.

Thank you very much for the feedback.

Final Decision Letter:

Dear Ms Padilla-Iglesias,

We are pleased to inform you that your Article "Deep history of cultural and linguistic evolution among Central African hunter-gatherers", has now been accepted for publication in Nature Human Behaviour.

Please note that *Nature Human Behaviour* is a Transformative Journal (TJ). Authors may publish their research with us through the traditional subscription access route or make their paper immediately open access through payment of an article-processing charge (APC). Authors will not be required to make a final decision about access to their article until it has been accepted. Find out more about Transformative Journals

We welcome the submission of potential cover material (including a short caption of around 40 words) related to your manuscript; suggestions should be sent to Nature Human Behaviour as electronic files (the image should be 300 dpi at 210 x 297 mm in either TIFF or JPEG format). Please note that such

pictures should be selected more for their aesthetic appeal than for their scientific content, and that colour images work better than black and white or grayscale images. Please do not try to design a cover with the Nature Human Behaviour logo etc., and please do not submit composites of images related to your work. I am sure you will understand that we cannot make any promise as to whether any of your suggestions might be selected for the cover of the journal.

With best regards,

[REDACTED]